

# Numerical simulations of glacier evolution performed using flow-line models of varying complexity

Antonija Rimac[1, 2], Sharon van Geffen[1, 2], and Johannes Oerlemans[1]

[1]Institute for Marine and Atmospheric Research, Utrecht University, The Netherlands
[2]Netherlands Earth System Science Centre, Utrecht University, The Netherlands

*Correspondence to:* Antonija Rimac (a.rimac@uu.nl)

**Abstract.** The performance of two numerical models of different complexity, i.e., a Shallow Ice Approximation (SIA) and a Full-Stokes Model (FSM), is studied by analyzing glacier evolutions at various bed geometries and by applying different climatic forcings. Glacier bed geometry changes from a constant slope and a uniform width to a superimposed Gaussian bump or ice-fall on a constant slope and an exponentially varying width. Constant slopes of 0.1, 0.2 and 0.3 are chosen to study the
evolution of a large, medium and small glacier, respectively. A specific mass balance serves as a climatic forcing. The steady state is reached 60, 30 and 10 years, respectively faster for large, medium and small glacier, when simulations are performed using SIA instead of FSM. Glaciers time response is studied by using step and periodic changes, and by imposing natural variability in the equilibrium-line altitude. Glacier length response time is up to 14 years longer when FSM is used compared to SIA. When periodic and natural variability are enforced, glaciers simulated using SIA lag in phase compared to the forcing
up to 81.2° for glacier length and up to 56.5° for volume. Contrary to that, glaciers simulated with FSM show greater lag in phase compared to the forcing for glacier length and smaller lag for volume. The models differ in their treatment of glacier flow mechanics and differences in physical variables become apparent with increasing glacier bed slope and bed profile complexity.

## 1 Introduction

Glaciers are generally thought to be good indicators of a global climate change on time scales of decades to centuries (Oerle-
mans, 1986). Records of glacier fluctuations and changes in climatic conditions can be used to project future changes in glacier size and spatial distribution (e.g., Oerlemans, 2001). For a correct simulation of glaciers, their bed characteristics and flow mechanics along with the spatial and temporal distribution of surface mass balance must be taken into account. Over the last few decades, glaciers have been the subject of many studies relying on different complexities of the used numerical models (e.g., LeMeur et al., 2004; Schäfer et al., 2008). But the performance of models of different complexity, and therefore the
applicability of these models to simulate future glacier volume and length evolution is still under debate (e.g., Leysinger-Vieli and Gudmundsson, 2004).

    The first simulations of glaciers were obtained by using the Shallow-Ice Approximation (SIA), which proved successful due to its potential to reduce the complexity of model equations and boundary conditions. The SIA has been used extensively to study the influence of changing climatic conditions on glacier volume and length evolution. Most of the previous studies

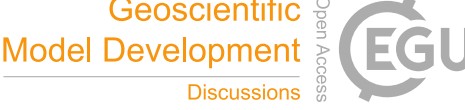



used a known glacier bed geometry, with a specific mass balance generated from climatological data serving as the forcing function (e.g., Oerlemans, 1986, 1997a, b). These studies proved that for smooth shaped glaciers SIA works well to model glacier fluctuations as a response to climatic forcing over time scales longer than a few years. This makes SIA a very efficient and lightweight alternative simulation tool to CPU and memory demands of more complex full-Stokes models (LeMeur et al.,

2004; Schäfer et al., 2008).

Several authors evaluated the performance of SIA and compared it to a complex full-system model. Raymond et al. (1989) used models of different complexity to calculate changes in glacier length induced by climate change. They concluded that the glacier response is a complex process requiring sophisticated full-system models. Pattyn (2002) calculated the response of Haut Glacier d'Arolla to perturbations in the ice thickness and mass balance using a full-system and SIA model. He concluded

that the non-linear behavior in the mass transport is responsible for a 10% difference in the time-lag of glacier response to the climatic signal in the full-system model compared to SIA. Further, Leysinger-Vieli and Gudmundsson (2004) studied the reaction of typically sized alpine glaciers using a full-system and an SIA model. Their comparison yielded no significant differences in glacier advance and retreat rates to shifts in the equilibrium-line altitude between the two models. Moreover, using a realistic height-mass balance feedback, only slight model-dependent changes in the steady state lengths were found.

LeMeur et al. (2004) and Schäfer et al. (2008) looked at the differences in physical variables, i.e., surface velocity and ice thickness, between their SIA and full-Stokes models. They concluded separately that the SIA surface velocity overestimates the full-Stokes model velocities by a factor 5 to 10.

These previous studies highlight large disagreements in simulations of glaciers in numerical models of different complexity and, at the same time, their time evolution is crudely studied. Here, we focus on the differences in glacier volume and length

evolutions simulated using two numerical models of different complexity. We systematically build up the complexity of the problem by applying several configurations of climatic forcing and glacier bed characteristics. The climatic forcing is imposed by means of variation in the equilibrium-line altitude $E$ in the mass balance equation. The bed characteristics are changed from a simple glacier bed with a constant slope and uniform width to a longitudinally varying bed slope and width. The applied models are quite different with respect to their mathematical formulation. The first model is a full-Stokes model that includes

stationary momentum equations, whereas the second model is based on the SIA, which ignores all stress gradients except for the vertical shear stress, and is a highly simplified but accurate description of the glacier flow that is not dominated by sliding.

This paper is organized as following: in the next section, we detail the model configurations used to simulate glacier length and volume evolutions, and we give an overview of the performed experiments. In Sect. 3.1 we describe glacier length and volume change for a constant bed slope and a constant equilibrium-line altitude. Based on these results, we build up our analysis

and introduce an instantaneous step (Sect. 3.2) and periodic change (Sect. 3.3) in the equilibrium-line altitude as a climatic forcing function. We study the effect of natural variability on a volume and length evolution in Sect. 3.4 to test the glacier reaction time. Furthermore, we address the influence of a longitudinally varying glacier width while retaining a constant bed slope in Sect. 4.2. In Sect. 4.3 the bed profile has a superposed Gaussian bump, and in Sect. 4.4, we introduce a steeper part, i.e. an ice-fall, and we apply a longitudinally varying glacier width. A summary and discussion is given in the final section.





## 2 Model description and experimental design

### 2.1 Numerical models

We simulate the transient evolution of an ice mass using two distinct ice flow models: a) a model based on the Shallow-Ice Approximation (hereafter SIA) (e.g., Oerlemans, 1986, 1997b), and b) a Full-Stokes Model (FSM) using the finite-element code Elmer/Ice (Gagliardini et al., 2013). SIA simulates glacier fluctuations well over time scales longer than a few years as shown by, e.g., Oerlemans (1986, 1997a, b). SIA is a one-dimensional (vertically integrated) flow line model. The vertically averaged internal deformation and sliding velocity are determined locally from the driving stress, which depends on the local surface slope and a ice thickness. The dynamic behaviour of the glacier is described in terms of changes in the glacier surface, which is calculated from the continuity equation (Oerlemans, 1986). In the model, a simple shearing flow and the Weertman type sliding law (Weertman, 1957) are considered separately. SIA takes into account the effect of geometry on volume conservation. The lateral geometry is parametrized by a trapezoidal cross section. The glacier width $W(x) = W_0(x) + \lambda H(x)$ is defined by the width at the bed $W_0(x)$, the local ice thickness $H(x)$ and the angle between the valley wall and vertical $\lambda(x) = 0.5x + 0.01$.

The numerical solution of the FSM equations is based on a finite-element code that has been used extensively for the ice modeling work (Gagliardini et al., 2013). The results from previous studies that use FSM based on the Elmer/Ice code are similar to ones from other tested full-Stokes models (Pattyn et al., 2008), giving a high confidence in the validity of the numerical solution. The equations in FSM are solved in three-dimensions, but in this study we use a vertical two-dimensional set-up along the flow line. In FSM, the velocities of ice at the free surface are a result of the Stokes solution. After this, the kinematic boundary condition for the free surface is applied to allow for the ice evolution. A Dirichlet boundary condition is applied on the upper surface, but only where the ice enters the domain, i.e., in the accumulation area (Gagliardini et al., 2013). At he lower boundary a Weertman type of friction law (Weertman, 1957) is applied. A classic isotropic form of Glen's flow law (Gagliardini and Meyssonnier, 1997) and solvers to compute deviatoric stress fields from the Stokes solution (Gagliardini et al., 2007) are also implemented. Our set-up for FSM uses a two-dimensional geometry, thus some three-dimensional processes such as mass divergence/convergence due to longitudinal gradients in the channel width are neglected. By assuming a trapezoidal channel shape (as set-up in SIA) and by calculating the channel width at the surface, we are able to estimate the flux contributions $Correction$ from changes in the channel width along the flow line (see Appendix):

$$Correction = -H \left( \frac{W_0 + \frac{1}{2}\lambda H}{W_0 + \lambda H} - 1 \right) \frac{\partial u}{\partial x} - \frac{uH}{W_0 + \lambda H} \frac{\partial W_0}{\partial x} - \frac{H^2 u}{2(W_0 + \lambda H)} \frac{\partial \lambda}{\partial x}, \quad (1)$$

where $H$ is the ice thickness, $W_0$ is the defined glacier width (see Eq. 8) and $u$ is the horizontal velocity. This is then applied as an additional mass balance term.

Before any experiments are carried out, we make sure that both SIA and FSM have the same initial and boundary conditions, and that they use same type of sliding law. Models are run for 500 years starting from zero glacier volume. The horizontal velocity is zero at the computational boundaries meaning that there is no inflow at the upper boundary. The model equations are solved on a structured grid of 100 m horizontal resolution. In FSM, we apply a vertical discretization that consists of 11 equally distributed nodal points. The time step of one year is used in both models for all experiments.

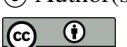



At the bedrock, we set up a Weertman-type of sliding law (Weertman, 1957). In SIA, the sliding law influences the vertically averaged ice velocity through sliding $f_s$ and deformation parameters $f_d$ (Budd et al., 1979) as:

$$
\begin{aligned}
U &= U_{d(SIA)} + U_{s(SIA)} \\
&= f_d H \tau_{d(SIA)}^3 + f_s \frac{\tau_{d(SIA)}^3}{H},
\end{aligned}
\tag{2}
$$

where $\tau_{d(SIA)}$ is the local driving stress and $H$ is the ice thickness. The values for deformation $f_d$ and sliding $f_s$ proposed by Budd et al. (1979) are $0.06 \cdot 10^{-15}$ Pa$^{-3}$ yr$^{-1}$ and $1800 \cdot 10^{-15}$ Pa$^{-3}$ m$^2$ yr$^{-1}$, respectively. As these values are empirical constants that depend on bed conditions, crystal structure and debris content (Oerlemans, 1986), they can be subject to some adjustments (e.g., Greuell, 1992). Thus, in Table 1 we present the prescribed sliding and deformation parameters that are tuned in order to simulate equal steady state length and volume (for three different glaciers) as the ones simulated using FSM during the initial spin-up phase. In Eq. 2, the driving stress $\tau_{d(SIA)}$ is defined with the ice density $\rho$, gravity $g$, the ice thickness $H$ and the surface elevation $h$, as:

$$
\tau_{d(SIA)} = -\rho g H \frac{\partial h}{\partial x}.
\tag{3}
$$

In FSM, the sliding law is specified through basal shear stress $\tau_{b(FSM)}$ defined as a non-linear function of a basal (sliding) velocity $U_{b(FSM)}$ as:

$$
\tau_{b(FSM)} = C U_{b(FSM)}^m,
\tag{4}
$$

where $C$ is the sliding parameter and $m = 1/3$ is the Weertman exponent. Again, in the initial experiment (presented in Sect. 3.1) we tune the sliding parameter to obtain the same glacier length and volume in the steady state to the ones from SIA (Table 1). The prescribed values for the sliding/deformation parameters are then kept constant for the experiments presented in Sect. 3.2 - 4.4. By analyzing the equations of the defined sliding laws, we can derive the possible sliding parameters in FSM based on the defined sliding parameter in SIA. If we assume that the sliding velocities are equal in the two models, and that the driving force in SIA is equal to the basal shear stress in FSM, i.e. $U_{s(SIA)} = U_{b(FSM)}$ and $\tau_{d(SIA)} = \tau_{b(FSM)}$, respectively, we can get a relation for the sliding parameter in FSM based on the one in SIA $f_s$:

$$
C = \left(\frac{H}{f_s}\right)^{1/3}.
\tag{5}
$$

If the mean ice thickness is assumed to be $H = 100$ m, the sliding parameter in FSM is about 0.038 MPa m$^{-1/3}$ yr$^{1/3}$, which is very close to our prescribed value for the large glacier.

## 2.2 Performed experiments

We perform several experiments to assess differences in glacier evolution between SIA and FSM. We study a simple glacier with a uniform width that rests on a bed with a constant slope of 0.1, 0.2 and 0.3. We choose for these different slopes to study the evolution of a *large*, *medium* and *small* glacier, respectively. Our numerical models are subject to the same forcing where





the mass balance serves as a climatic forcing function. It is specified as a linear function of surface elevation $h$ and it is defined in m water depth $\text{yr}^{-1}$ as:

$$M = \beta(h - E), \tag{6}$$

where $\beta = 0.008$ m.w.e $\text{yr}^{-1}$ $\text{m}^{-1}$ is a characteristic balance gradient for mountain glaciers (Lüthi and Bauder, 2010) and $E$

is the equilibrium-line altitude set to 1700 m for the large glacier and 1600 m for the medium and small glaciers. For the large glacier, we choose for the greater height of $E$ in order to prevent the glacier from reaching the end of the computational domain which is at a length of 10000 m, while for the smaller glaciers we choose for the lower height to allow the glaciers to develop.

Second, after the initial spin up of 500 years, we study the response time of glaciers in our models to stepwise forcing. Every 500 years $E$ is instantaneously changed by 60 m. Third, we study the influence of periodic fluctuations in $E$ after the initial

simulation of 500 years. Here, $E$ varies periodically as:

$$E(t) = E + 60\sin(2\pi t/T), \tag{7}$$

where $t$ is time variable in years, $T$ is the period of the forcing, and the amplitude is 60 m for all glaciers. The period $T$ is changed to analyze the influence of slowly (by 1000 years), intermediately (500 years) and fastly fluctuating climate (200 years).

Lastly, we are interested in glacier fluctuations on a decadal-to-centennial time scale. Therefore, we simulate glacier volume and length evolution under a small period random forcing and under an observed temperature time series. In the former, we simulate the inter-annual variability by imposing uniform random noise with the mean equilibrium-line altitude $E$ of 1700 m and with the amplitude of the variability of $\pm$ 100 m in the mass balance equation. In the latter, the used forcing function is based on the Central European annual temperature variability (Luterbacher et al., 2004). The forcing is obtained by converting

the temperature variability to equilibrium-line variability with a factor of 100 m/°C.

To address the difference in physical variables of glaciers simulated using SIA and FSM, we apply several configurations of glacier bed characteristics. First, as already mentioned, we study a simple glacier with a uniform width that rests on a bed with a constant slope. Second, we assume that the glacier width changes exponentially along the flow line as:

$$W_0(x) = w_0 + w_1 x e^{-ax}. \tag{8}$$

Here, $w_0 = 1000$ m denotes the width at the terminus, while $w_1 = 3$ and $a = 0.009$ $\text{m}^{-1}$ shape the glacier.

Third, we apply a bed with a reversed slope over a certain distance along the flow line, i.e., the bed profile is obtained by superposing a Gaussian bump on a linearly sloping bed as:

$$b(x) = b_0 - sx + b_1 \cdot e^{-[(x-x_0)/x_1]^2}, \tag{9}$$

where $x$ is a distance from glacier head in m, $b_0 = 2000$ m is the height at the glacier head, $s$ is the applied bed slope that

represents the linear part of the bed profile, $b_1 = 250$ m is the amplitude of the bed bump, $x_0 = 3000$ m is the location and $x_1 = 500$ m is the width of the bump along the x-axis.




Lastly, we analyze model simulations with the glacier bed characterized by a steeper part, i.e., an ice-fall:

$$
b(x) = \begin{cases} b_0 - sx & \text{if } x \leq 3000\,m \\ b_0 - sx - s_1 x & \text{if } 3000\,m < x \leq 5000\,m \\ a_0 - sx & \text{if } x > 5000\,m. \end{cases} \tag{10}
$$

The constant slopes $s$ of 0.1, 0.2 and 0.3 are already defined to allow the evolution of the large, medium and small glacier, respectively. The slope $s_1$, that defines the ice-fall, is 0.2 for the large, 0.3 for the medium and 0.4 for the small glacier. The bed height after the applied ice-fall is $a_0$. As in the second experiment, in the last two experiments we assume that the glacier width changes exponentially along the flow line.

## 3 Influence of climatic forcing on glacier volume and length evolution

In this section, we analyze the glacier volume and length evolution in SIA and FSM under different climatic conditions. The purpose of this is to study the sensitivity and response of glaciers simulated with our models to a wide range of climatic forcings. After the equilibrium states of three different glaciers resting on a bed with a constant slope and a uniform width in our models are reached, we study the dynamical aspects in which glaciers gradually adjust their size to changing environmental conditions, i.e., the equilibrium-line altitude.

### 3.1 A linear mass balance profile

We study glacier volume and length evolution (Fig. 1a and 1b, respectively) over 500 years of simulation starting from zero ice volume and with the specific mass balance as given in Eq. 6. The results are obtained using SIA (solid line) and FSM (dashed line) for glaciers on bed slopes of 0.1 (large glacier, red), 0.2 (medium glacier, blue) and 0.3 (small glacier, green). For better clarity, the glacier length is shifted by $\pm 1$ km along the y-axis for the large and small glacier.

Figures 1a and 1b show that after some initial time in the calculation (about 60 years for large, 30 years for medium and 20 years for small glacier), glacier volume and length calculated using SIA are larger than those calculated using FSM. As the time proceeds and the glaciers approach a steady state, these differences become progressively smaller, consistent with Leysinger-Vieli and Gudmundsson (2004). For large, medium and small glaciers, the steady state is reached about 60, 30 and 10 years, respectively, faster in SIA compared to FSM. The steady state glacier lengths are equal in both models and glacier volumes differ by less than 0.4%. This equality was a prerequisite for the following experiments in which we investigate glacier response to changes in $E$. To address the model dependent differences in response to climate variability accurately, we have to start from the same steady state lengths and volumes, and use the same forcing function.

### 3.2 Step change in the equilibrium-line altitude

By making an instantaneous step change in the equilibrium-line altitude $E$, the glaciers are forced to advance or retreat. In this experiment, every 500 years $E$ is changed by 60 m for all glaciers. The calculated changes in volume and length obtained





using SIA (blue) and FSM (red) are shown in Fig. 2 for the large glacier. The volume and length evolution is similar for all glaciers, thus, for better clarity we present the result only for the large glacier. Change in $E$ is shown with a black line. The result indicates that the step change in $E$ causes a gradual adaptation of the glacier to a new climatic forcing. This reaction is almost an exponential change of the glacier volume and length in both models. Following the evolution of the large glacier

for both length and volume we can see that the rate of advance and retreat differs in our models with SIA producing shorter advance and retreat compared to FSM. Here, the inclusion of horizontal stresses leads to elevation changes that, along with the height-mass balance, causes differences in the advance and retreat rates (Leysinger-Vieli and Gudmundsson, 2004).

In this study, we make reference to an e-folding response time for glacier volume $\tau_V$ and length $\tau_L$ following Oerlemans (1997a):

$$
\begin{aligned}
\tau_V &= t\Big(V = V_2 - \frac{V_2 - V_1}{e}\Big) \\
\tau_L &= t\Big(L = L_2 - \frac{L_2 - L_1}{e}\Big).
\end{aligned}
\tag{11}
$$

Here, $V_2$, $V_1$, $L_2$ and $L_1$ are the new and old steady state glacier volumes and lengths, respectively. The steady states are affected by two climatic conditions that depend on the instantaneous change in $E$ acting as a forcing function. In Table 2, the e-folding volume and length response times are listed for each glacier individually, and for both SIA and FSM. In accordance with the previous study by e.g., Oerlemans (1997a), the results indicate that the volume response time is shorter than the length

response time because volume is more directly affected by the climate change (Oerlemans, 1997a). Also, the response time for both glacier length and volume reduces with increasing bed slope, since larger glaciers resting on smaller bed slopes have stronger height-mass balance feedback, which in turn increases the glacier response time (Oerlemans, 2001). The differences between SIA and FSM are also notable, with SIA adjusting faster to the changing climate than FSM. Unlike in Leysinger-Vieli and Gudmundsson (2004), whose full-Stokes model did not show consistent lag or lead in comparison to their SIA model,

here, the response time in FSM is longer than the response time in SIA for all glaciers.

### 3.3 Periodic change in the equilibrium-line altitude

We impose a periodic forcing to examine time-dependent solutions of our numerical models. The periodic forcing is applied after the initial simulation of 500 years. Here, the specific mass balance does not only linearly depend on the surface elevation, but it also includes a periodic variation in $E$ (Eq. 7). The responses of glacier length and volume to the periodic forcing are

shown in Fig. 3 for SIA (blue) and FSM (red). The result is presented only for the large glacier for better recognition and the same conclusion can be reached for smaller glaciers. Figure 3 shows the results for a forcing with a period of 200 years. We can see that both models adjust well to the new forcing. The glacier volume again responds faster to the change in climatic forcing compared to glacier length. Figure 3b indicates that the glacier length in SIA reacts faster to the new forcing compared to length in FSM. The opposite is true for glacier volume that reacts faster in FSM compared to SIA. Also, as already seen, it

is notable that glacier volume and length amplitudes are not equal in the two models.

To estimate the reaction of the models to the periodic change in $E$, we calculate the phase difference between the glacier response and the climatic forcing (Table 3). The phase difference calculation is based on a Fourier transform of a periodic





change in glacier volume and length. The results indicate that the phase lag between the forcing and the model response typically increases with increasing size of the glacier as a result of height-mass balance feedback Oerlemans (2001). Also, the phase lag increases with increasing frequency of the climatic forcing, possibly as a result of grid discretization since when a grid resolution is increased from 100 m to 20 m the phase lag decreases for almost 50%. In general, the phase difference for

glacier length is higher by up to 8° for the large glacier forced at a period of 200 years for FSM than for SIA. For glacier volume, the phase difference is higher in SIA by up to 10° for the small glacier forced at a period of 200 years. This indicates that the glacier length in SIA increases faster, possibly because of the higher sliding velocity in SIA compared to FSM, which, in turn, makes the volume response in SIA slower.

### 3.4 Year-to-year fluctuating mass balance profile and annual variation in the surface air temperature

To investigate the reaction of glaciers to year-to-year variability in the equilibrium-line altitude $E$ in the mass balance equation (Eq. 6) we apply two forcing functions. First, a uniform noise with a standard deviation of 100 m is superposed on $E$. Second, a forcing function derived form the Central European mean annual temperature record (Luterbacher et al., 2004) is used. Here, the derived time series have a length of 505 years.

In Fig. 4, we show volume and length evolution obtained from model simulations with year-to-year varying mass balance,

i.e., the uniform noise (left panel), and the annual temperature variability (right panel) for SIA (blue) and FSM (red) for the large glacier. The results of both simulations are alike in the two models, having a correlation of more than 99%. But the obvious difference, which can be seen in glacier length evolution (Fig. 4b and 4d), is that FSM always lags SIA for glacier advance. For glacier retreat, the length decreases simultaneously in the two models. This is not the case for glacier volume evolution, where SIA lags FSM (see e.g., years 760-840 on Fig. 4d). Additionally, it is notable that glacier length evolution in

FSM does not respond to some smaller short-term variability (e.g., year 800-900 and 1100-1200 on Fig. 4b). Figures 4c and 4d indicate a periodic reaction of glacier length and volume to year-to-year variability in the temperature record. The power spectrum (not shown) also shows high energy at a frequency of 250 years that seems to reflect on the glacier evolution.

## 4 Influence of bed characteristics on glacier physical properties

To see how different physical complexities of our models influence the results, we study the steady state surface elevation

profile, vertically-averaged horizontal ice velocity and basal shear stress. Additionally, we compare components of a force balance equation to gain a better insight into the forces that are important for the particular glacier or part of the glacier. Following van der Veen (1999), the force balance equation integrated over a cross section is given as:

$$\underbrace{\frac{\partial}{\partial x}(2HW\overline{\tau_{xx}})}_{\text{I}} + \underbrace{(\tau_{y2} + \tau_{y1})H}_{\text{II}} + \underbrace{\tau_b W}_{\text{III}} = \underbrace{-\rho g H \frac{\partial h}{\partial x} W}_{\text{IV}}, \tag{12}$$

where term I represents the longitudinal gradient of the vertically averaged normal stress $\overline{\tau_{xx}}$, term II formulates the side drag,

but it is excluded in the current model set-up, term III is the basal drag and term IV is the driving force. In Sect. 4.1 the glacier



width $W$ is kept uniform and it can be excluded from the equation, but in Sect. 4.2 - 4.4, the longitudinally varying glacier width is adopted. Simulated force balance components are only compared for FSM because in SIA the driving force is fully balanced by the basal shear stress.

### 4.1 Constant bed slope and exponential change in glacier bed width

To see the influence of the opposing physical complexity of SIA and FSM on our results, we study the steady state surface elevation profile (Fig. 5a, only for the large glacier), vertically averaged horizontal ice velocity (Fig. 5b) and the shear stress (Fig. 5c). In Fig. 5d, the steady state result for the driving force is given in green, the basal drag in blue and the horizontal gradient of the normal stress is given in red. The result is shown for the large glacier only, because the results are similar for the other two glaciers.

Surface elevation profiles shown in Fig. 5a are similar in SIA and FSM with a mean ice thickness of about 104 m. Two small differences appear at the glacier boundaries due to the imposed boundary condition at the glacier head in SIA as already mentioned, and due to different numerical differencing scheme used to evaluate time evolution of ice height at the glacier terminus (Oerlemans, 2001).

The vertically-averaged ice velocity in SIA (Fig. 5b) depends on the sliding and deformation parameters, the ice thickness
and the driving stress, as introduced in Sect. 2 (Eq. 2). On the other hand, in FSM, the velocity field of an ice mass flowing under gravity is obtained by solving the Stokes equations combined with the sliding law at the basal boundary. This sliding law then represents a relation between the basal shear stress and the sliding velocity. Because of these different definitions of the ice velocity, some differences in the results between the two models are expected. For the large glacier, velocities are most alike in the two models. For smaller glaciers, the differences become more obvious. This difference arises from the sliding law.
Increasing the bed slope reduces the glacier length and volume (e.g., Weertman, 1961; Oerlemans, 1980; Hyde and Peltier, 1986; Oerlemans, 2001). Thus, in order to reach equal steady state volume and length in the two models we have to introduce lower sliding parameters which finally leads to higher velocity in FSM. For the medium and small glaciers, we would reach equal steady state velocities in the two models by increasing the sliding parameter in FSM, but at the cost of losing the equal steady state glacier volume and length (not shown).

The basal shear stress in SIA, which is proportional to the ice thickness and the bed slope (Eq. 3), increases with the glacier ice thickness and glacier length (Fig. 5c). In FSM, on the other hand, the basal shear stress is calculated from the sliding velocity and the Weertman sliding parameter (Eq. 4). It is expected that increasing the velocity with increasing slope would lead to higher basal shear stress. However, the decreasing sliding coefficient with the bed slope plays a more prominent role on the basal shear stress in FSM. Differences between the models at the glacier boundaries (Fig. 5c) are a result of numerical
instabilities in FSM with a wave length of 400 m. The mentioned instabilities have a very analogous behaviour to the grounding line problem, and these stress oscillations have some physical origin in a stress-boundary layer produced by a sudden change in bedrock stress conditions (Thomas Zwinger, personal communication). By increasing the horizontal spatial resolution of the simulation from 100 to 20 m, the amplitude of this instability reduces by 30% while the wave length becomes 80 m.





Figure 5d shows an almost perfect balance between the driving force and the basal shear stress. As the bed slope increases for smaller glaciers (not shown), compression and extension become more pronounced, i.e. the longitudinal normal stress gradient increases, and partly balances the driving stress (Eq. 12). In the absence of the normal stress gradient in the SIA model, the driving force (in absence of lateral drag) is completely balanced by the basal drag.

## 4.2 Constant bed slope and exponential change in glacier width

By introducing a spatially-varying bed width, we increase the complexity of our numerical models and their governing equations. The influence of varying bed width on the glacier may create interesting dynamic behavior. Here, we analyze the steady state surface elevation profile (Fig. 6a, only for the large glacier) simulated with SIA (solid line) and FSM (dashed line). The steady state vertically-averaged horizontal ice velocity (Fig. 6b) and basal shear stress (Fig. 6c) are studied for all three glaciers (large glacier in red, medium glacier in blue and small glacier in green). We also compare the force balance components, but again only for the large glacier (Fig. 6d).

The steady state surface elevation profile (Fig. 6a) is similar in both models. Apart from the first 2000 m from the glacier head, the ice thickness simulated using FSM is about 10 m higher compared to the one simulated using SIA. The vertically-averaged ice velocity is similar in both models for all three glaciers (Fig. 6b). Here, the correction term accounting for the flow convergence/divergence added to the mass balance equation in FSM (Eq. 1), leads to a lower velocity for smaller glaciers and downward shift of the velocity maximum, compared to experiments performed in Sect. 4.1. The basal shear stress (Fig. 6c) reduces with increasing bed slope in both models, and again the difference between the two models is more pronounced for smaller glaciers. Here, and in the following sections, the force balance components shown in Fig. 6d include the varying bed width. Applying this varying width leads to increase in the normal stress (compression) where the width decreases, and to decrease in normal stress (extension) where the width increases.

## 4.3 Gaussian bump superposed on linearly sloping bed and exponential change in glacier width

Following the previous section, we further increase the complexity of our glacier bed characteristics by superposing a Gaussian bump on a linearly sloping bed (Eq. 9). We also assign a non-uniform glacier width (Eq. 8). In Fig. 7, we show the surface elevation profile, vertically-averaged horizontal ice velocity and the basal shear stress for both SIA (solid line) and FSM (dashed line), and we compare the force balance components simulated using FSM. Here, and in the following section we plot the results for the large glacier only, because the results are similar for all three glaciers.

The glacier steady state surface elevation profile, seen in Fig. 7a, is similar in both models. Some differences appear at the glacier head due to the difference in imposed boundary conditions, at the bump possibly because of the difference in physics between the two models (in FSM, the normal stress gradient is considered, and this force is significant around the bed bump as it affects the basal shear stress which is directly related to the basal velocity), and at the lower part of the glacier possibly because the glacier undergoes more sliding and deformation in SIA compared to FSM as seen in Schäfer et al. (2008).

The horizontal glacier ice velocity (Fig. 7b) is similar in both models with an exception at the downstream side of the bump. There, the driving stress is large because of the large surface slope. Since in SIA the driving stress is balanced solely by the



basal shear stress, the basal shear stress and therefore the velocity become large at the bump's downstream. In FSM, the driving stress is partly balanced by the positive normal stress gradient (Fig. 7d). This compression leads to increase in the ice velocity but it is not so pronounced as in SIA. The basal shear stress (Fig. 7c) in SIA increases with increasing ice thickness. Upstream of the bed bump, the driving stress, and thus the basal shear stress, increases due to the gradual increase in the ice thickness. At

the length of about 3000 m, or just upstream of the bed bump, the ice thickness decreases, but the surface slope increase leads to a further increase in the driving stress. Above and just downstream of the bump, the driving stress reaches its maximum due to the large surface slope. FSM gives a different result. The initial increase is smaller than in SIA. Here, the longitudinal stress gradient is about zero, so in FSM the driving stress is balanced by the basal shear (Fig. 7d). Smaller ice thickness and surface slope in the FSM upstream of the bed bump, eventually lead to the smaller basal drag.

## 4.4 Ice-fall imposed on the linearly slopping bed and exponential change in glacier width

Many glaciers are characterized by a rapid flow caused as a result of an ice-fall that occurs at places where the glacier bed steepens. After the ice-fall, the glacier bed flattens and the ice flow slows down. In this section, we induce the ice-fall on a linearly sloping bed (Eq. 10), and we again assume to have a varying glacier width. As in Fig. 5 - 7, in Fig. 8 we present the steady state surface elevation profile, vertically-averaged horizontal ice velocity and basal shear stress. The results are obtained

using SIA (solid line) and FSM (dashed line), and are again presented only for the large glacier. Additionally, we compare the force balance components simulated using FSM.

The surface elevation profile (Fig. 8a) is similar in both models. Some minor exceptions that appear at glacier margins and at the ice-fall are already discussed in previous sections. The horizontal ice velocity (Fig. 8b) increases at the ice-fall but the increase is again more pronounced in SIA compared to FSM. This velocity increase, when a fast changing glacier bed profile

is used, is influenced by the lack of normal stress in SIA. The basal shear stress is higher in SIA compared to FSM (Fig. 8c), and there is again a notable numerical instability at the glacier terminus in FSM. The basal shear stress and the driving force in the force balance equation (Fig. 8d), outside of the ice-fall area, are balanced, and the normal stress has only a significant influence at the ice-fall.

## 5 Discussion and conclusions

In this study, we present glacier volume and length evolutions based on numerical simulations performed with two flow-line models of different complexity. The performance of a simple one-dimensional Shallow-Ice Approximation model (SIA) is compared to the Full-Stokes Model (FSM) using the finite-element code Elmer/Ice. The surface mass balance, which is a linear function of the surface elevation, acts as a forcing of the models. The glacier bed characteristics change from a bed with a constant slope of 0.1, 0.2 and 0.3, and with a uniform width, to a bed with a superimposed Gaussian bump or ice-

fall on the constant slope and with an exponentially varying width. The mentioned slopes are used to simulate the evolution of a large, medium and small glacier, respectively. The two models differ in their physics, but are forced under the same



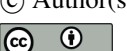

climatic conditions, they have the same initial and boundary conditions, and they have the same Weertman-type sliding law. We conclude the following:

- Glaciers in the two models do not develop identically. The steady state in SIA is reached 60, 30 and 10 years respectively faster compared to FSM for the large, medium and small glacier, respectively.

- By enforcing step and periodic change, and natural variability in the equilibrium-line altitude $E$, and by applying annual variation in the surface air temperature, we study response of our glaciers to changes in climatic conditions. In general, the e-folding response time of our glaciers is shorter in SIA compared to FSM by up to 4 years for glacier volume and up to 14 years for glacier length. Periodic change in $E$ leads to a phase difference between the glacier response and the forcing function. Here, SIA compared to FSM, has a longer lag in phase for glacier volume ($56.5°$ compared to $53.1°$ for the large glacier forced at a period of 200 years) and shorter lag for glacier length ($81.2°$ compared to $89.4°$ for the large glacier forced at a period of 200 years). The experiments using natural variability in $E$ and annual mean temperature confirm the finding that glacier length responds faster in SIA whereas glacier volume responds faster in FSM.

- Since the two models differ in their treatment of the mechanics of glacier flow, differences in the steady state surface elevation profiles, basal shear stress and ice velocity are expected. The surface elevation profiles are similar in both models. The differences at the glacier margins become more apparent when a complex bed profile, with varying surface slope and width, is included. The differences in the shear stress and ice velocity increase with increasing bed slope and with increasing bed profile complexity. The force balance components reveal that for the constant bed slope, the driving force is well balanced by the basal shear stress. When the bed slope and glacier width vary longitudinally, the normal stress gradient increases at places of the Gaussian bump and ice-fall.

Difference in physical properties between the two models, as seen, are reflected on e.g., the simulated basal shear stress and vertically-averaged horizontal ice velocity. This implies that the sliding and deformation velocity will be different in the two models (Fig. 9). In FSM, the sliding velocity (red dashed line) is calculated directly from the Navier-Stokes equations, whereas in SIA (red solid line) it is calculated from the driving stress. In general, the sliding and deformation velocity have similar profiles in both models. For a constant bed slope and a uniform glacier width, the sliding velocity is higher in SIA compared to FSM in the area of high ablation. This supports our conclusion that in SIA glaciers flow faster to the area of higher ablation, which in turn makes the volume response in SIA slower compared to FSM. When exponentially varying glacier width is included, FSM simulates slightly stronger (about 6 m yr$^{-1}$ higher) sliding and weaker deformation than SIA. This might appear as a result of the correction term (Eq. 1) accounting to the flow convergence/divergence added to the mass balance in FSM. At the bump and ice-fall, SIA simulates higher sliding velocity than FSM. There the high driving stress and low ice thickness lead to high sliding velocity in SIA. This does not necessarily mean that the highest surface velocity simulated using SIA of about 250 m year$^{-1}$ is overestimated. Wangensteen et al. (2006) showed that the highest surface velocity measured on Nigardsbreen (Norway) can reach 489 m year$^{-1}$ at the main ice-fall, which is almost twice as high than the highest velocity obtained in this study.





Experiments where the natural variability in $E$ and the annual variation in the surface air temperature are used as the forcing function indicate that both models adjust well to the new forcing with the correlation of up to 99%. But, glacier length evolution shows that FSM never reaches the maximum length of SIA for glacier advance, and it has lower minimum for glacier retreat. The difference between glacier maximum length in the two models is up to 200 m. This raises a question of which model

correctly simulates advance and retreat rates with the simplifications we have applied in this study..

In the similar experiments, Leysinger-Vieli and Gudmundsson (2004) showed that for the same absolute shift in the equilibrium-line altitude $E$, calculated glacier length and volume differences between their SIA and full-Stokes model are obtained only for glaciers shorter than 5 km. They showed that an SIA model produces a larger advance and a shorter retreat when $E$ respectively drops and rises. In the present study, however, for the equal drop or rise in $E$, no matter what length, glaciers almost always

show different steady state lengths in the two models. This advance and retreat are more pronounced for medium and small glaciers in SIA, while for large glacier the change is more pronounced when using FSM. Most importantly, Leysinger-Vieli and Gudmundsson (2004) could not find a systematic lag or lead of their SIA model compared to their full-Stokes model, and they only found some small differences in advance and retreat rates. Here, on the other hand, FSM constantly lags behind SIA, and SIA shows retreat rate that is 14 and FSM 25 years shorter for the large glacier.

The present results, in particular the simulated basal shear stress and horizontal velocity profiles, lead to some expected differences between the two models, since they are closely linked to model equations and their underlying assumptions. Moreover, our results show different glacier length and volume evolutions in the two models, where it takes at least 10 years longer time for FSM to reach the steady state compared to SIA. Also, glacier length simulated using FSM has longer response time compared to the one simulated using SIA. This indicates the need for further studies based on true climatic conditions and

actual bed profiles of existing glaciers to investigate how accurately models of different complexity can simulate observed glacier length evolution.

## 6 Code and data availability

Elmer/Ice is open source finite element software for ice sheet, glaciers and ice flow modelling freely available at a website: http://elmerice.elmerfem.org. The first author will provide SIA model code and the data produced by both models upon request.

Numerical computations were performed, and the data used for the analyses are archived at the IMAU computing server.

## Appendix A: Correction term for FSM

To calculate the flux that arises from the lateral spreading of ice, we start with a flux $q$ through a cross-section:

$$q = -Au, \tag{A1}$$





where $A$ is the cross-sectional area and $u$ is the horizontal velocity along the flow-line. We further assume a trapezoidal channel shape where the width at the surface $W(x,z)$ and an area consumed by ice $A(x)$ are:

$$W(x,z) \quad = \quad W_0(x) + \lambda z, \tag{A2}$$

$$\int_{h_b}^{h_t} W(x,z,t)\,dz \quad = \quad W_0(x)(h_t - h_b) + \frac{1}{2}\lambda(h_t - h_b)^2, \tag{A3}$$

thus

$$A \quad \equiv \quad W_0(x)(h_t - h_b) + \tfrac{1}{2}\lambda(h_t - h_b)^2, \tag{A4}$$

$$H \quad \equiv \quad h_t - h_b. \tag{A5}$$

$$\tag{A6}$$

10 Here, $\lambda$ is the angle between the valley wall and the vertical, $h_t$ is a total surface elevation and $h_b$ is the initial surface elevation. The mass conservation equation now yields a time derivative of the area $A$ as:

$$\frac{\partial A}{\partial t} \quad = \quad -\frac{\partial uA}{\partial x} \tag{A7}$$

$$(W_0 + \lambda H)\frac{\partial H}{\partial t} \quad = \quad -\left(W_0 H + \tfrac{1}{2}\lambda H^2\right)\frac{\partial u}{\partial x} - u(W_0 + \lambda H)\frac{\partial H}{\partial x} - uH\frac{\partial W_0}{\partial x} \tag{A8}$$

$$\frac{\partial H}{\partial t} \quad = \quad -\frac{W_0 H + \tfrac{1}{2}\lambda H^2}{W_0 + \lambda H}\frac{\partial u}{\partial x} - u\frac{\partial H}{\partial x} - \frac{uH}{W_0 + \lambda H}\frac{\partial W_0}{\partial x} \tag{A9}$$

15 $$\quad = \quad -H\frac{\partial u}{\partial x} - u\frac{\partial H}{\partial x} - H\left(\frac{W_0 + \tfrac{1}{2}\lambda H}{W_0 + \lambda H} - 1\right)\frac{\partial u}{\partial x} - \frac{uH}{W_0 + \lambda H}\frac{\partial W_0}{\partial x} \tag{A10}$$

$$\quad = \quad \underbrace{-\frac{\partial uH}{\partial x}}_{\text{FSM}} \underbrace{-H\left(\frac{W_0 + \tfrac{1}{2}\lambda H}{W_0 + \lambda H} - 1\right)\frac{\partial u}{\partial x} - \frac{uH}{W_0 + \lambda H}\frac{\partial W_0}{\partial x}}_{\text{Correction}} \tag{A11}$$

If we assume that $\lambda$ is a function of the longitudinal coordinate $x$, as it is the case in this study, the equation becomes:

$$\frac{\partial H}{\partial t} = \underbrace{-\frac{\partial uH}{\partial x}}_{\text{FSM}} \underbrace{-H\left(\frac{W_0 + \tfrac{1}{2}\lambda H}{W_0 + \lambda H} - 1\right)\frac{\partial u}{\partial x} - \frac{uH}{W_0 + \lambda H}\frac{\partial W_0}{\partial x} - \frac{H^2 u}{2(W_0 + \lambda H)}\frac{\partial \lambda}{\partial x}}_{\text{Correction}} \tag{A12}$$

*Competing interests.* The authors declare that they have no conflict of interest.

20 *Acknowledgements.* This work was carried out under the program of the Netherlands Earth System Science Centre (NESSC), financially supported by the Ministry of Education, Culture and Science (OCW). We thank Sue Cook, Thomas Zwinger and Olivier Gagliardini on their valuable input and discussion during the preparation of the manuscript. Numerical computations were performed, and the data used for the analyses are archived at the IMAU computing server.



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

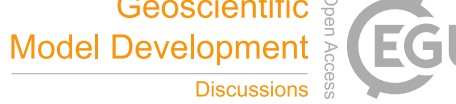



**Table 1.** Sliding and deformation parameters used to obtain equal steady state length and volume for the large, medium and small glacier is SIA and FSM during the initial 500 years long run.

|  | Large | Medium | Small |
|---|---|---|---|
| sliding in SIA $[*10^{-15}Pa^{-3}m^2yr^{-1}]$ : | 1646.4 | 1713.94 | 1705.953 |
| deformation in SIA $[*10^{-15}Pa^{-3}m^2yr^{-1}]$: | 0.084 | 0.085 | 0.057 |
| sliding in FSM $[MPa\, m^{-1/3}yr^{1/3}]$: | 0.043 | 0.03 | 0.0223 |

**Table 2.** Numerical volume $\tau_V$ and length $\tau_L$ response time to a sudden change in the equilibrium line altitude. The calculation is obtained for the large, medium and small glacier using SIA and FSM. The result is presented in years.

|  | Large | Medium | Small |
|---|---|---|---|
| $\tau_V$ in SIA: | 25 | 20 | 13 |
| $\tau_V$ in FSM: | 29 | 22 | 13 |
| $\tau_L$ in SIA: | 52 | 21 | 14 |
| $\tau_L$ in FSM: | 66 | 29 | 18 |

**Table 3.** Phase Difference (PD) between sinusoidal forcing and the glacier volume and length evolution obtained using SIA and FSM. Periodic change $T$ in the equilibrium-line altitude of 1000, 500 and 200 years (yr) is used. The calculation is done for large, medium and small glacier. The result is presented in degrees.

|  |  | Large | Medium | Small |
|---|---|---|---|---|
|  | $T$ of 1000 yr | 13.3 | 6.1 | 4.1 |
| PD Volume in SIA: | $T$ of 500 yr | 25.5 | 12 | 7.9 |
|  | $T$ of 200 yr | 56.5 | 28.6 | 18.3 |
|  | $T$ of 1000 yr | 12.7 | 4.9 | 2.8 |
| PD Volume in FSM: | $T$ of 500 yr | 24 | 9.2 | 3.4 |
|  | $T$ of 200 yr | 53.1 | 22.6 | 8.9 |
|  | $T$ of 1000 yr | 17.9 | 8.7 | 6.1 |
| PD Length in SIA: | $T$ of 500 yr | 35.1 | 17.7 | 11.7 |
|  | $T$ of 200 yr | 81.2 | 42.3 | 27.6 |
|  | $T$ of 1000 yr | 20.6 | 11.8 | 10.6 |
| PD Length in FSM: | $T$ of 500 yr | 38.6 | 20.2 | 13.5 |
|  | $T$ of 200 yr | 89.4 | 45 | 26.6 |



**Figure 1.** Simulation of glacier (a) volume and (b) length change obtained using SIA (solid line) and FSM (dashed line) for the large (red), medium (blue) and small glacier (green) configuration. The glaciers rest on a bed with a constant slope and a uniform width. Note that glacier length for large and small glaciers is shifted by ± 1 km along the y-axis for better recognition.





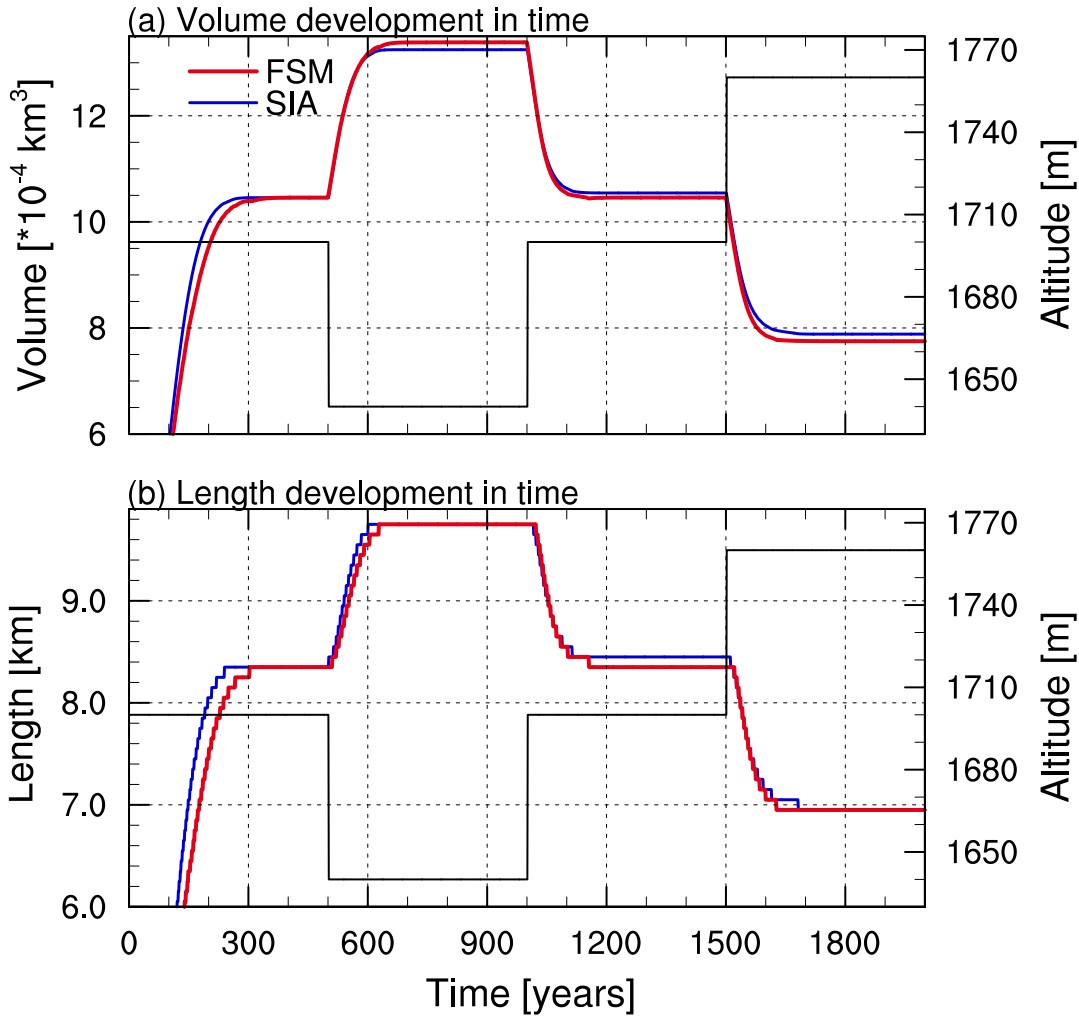

**Figure 2.** Simulation of glacier (a) volume and (b) length response to a sudden change in the equilibrium line altitude obtained using SIA (blue) and FSM (red) for the large glacier configuration. Black line represents a change in the equilibrium-line altitude. The glacier rests on a bed with a constant slope and a uniform width.





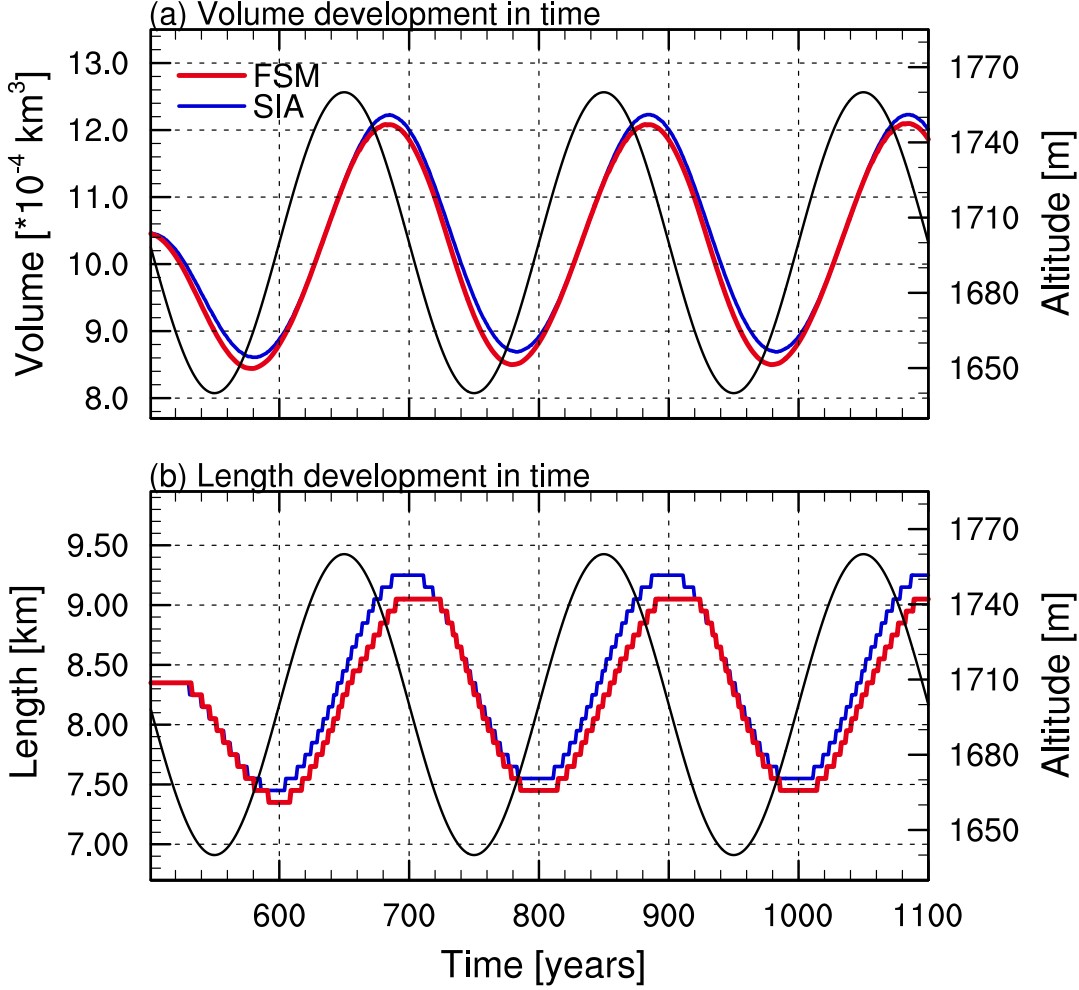

**Figure 3.** Simulation of glacier (a) volume and (b) length response to a periodic change in the equilibrium line altitude obtained using SIA (blue) and FSM (red) for the large glacier. Black line represents a change in the equilibrium-line altitude. Note that the the scale of the equilibrium-line altitude is reversed. The glacier rests on a bed with a constant slope and a uniform width.





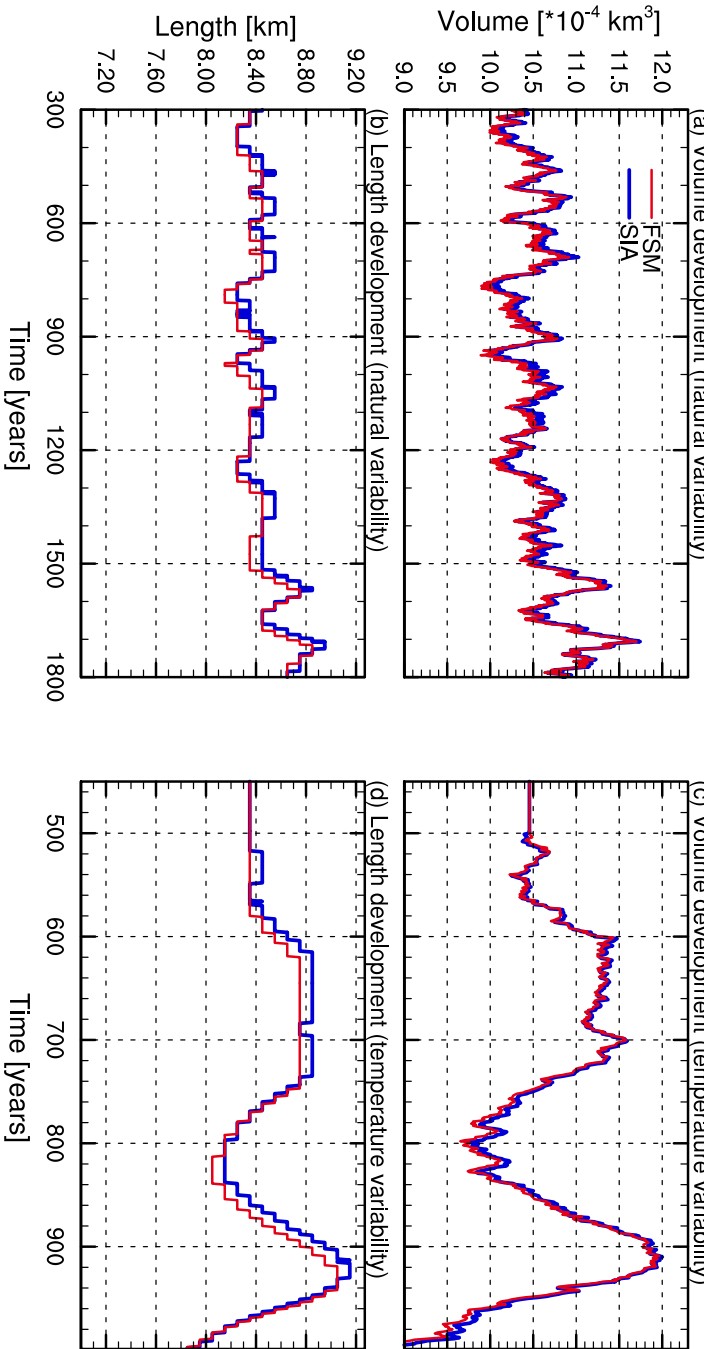

**Figure 4.** Simulation of glacier (a) volume and (b) length response to a natural change in the equilibrium-line altitude obtained using SIA (blue) and FSM (red) for the large glacier configuration. Simulation of glacier (c) volume and (d) length response to a change in the annual surface air temperature obtained using SIA (blue) and FSM (red) for the large glacier configuration. The glacier rests on a bed with a constant slope and a uniform width.

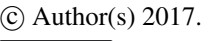



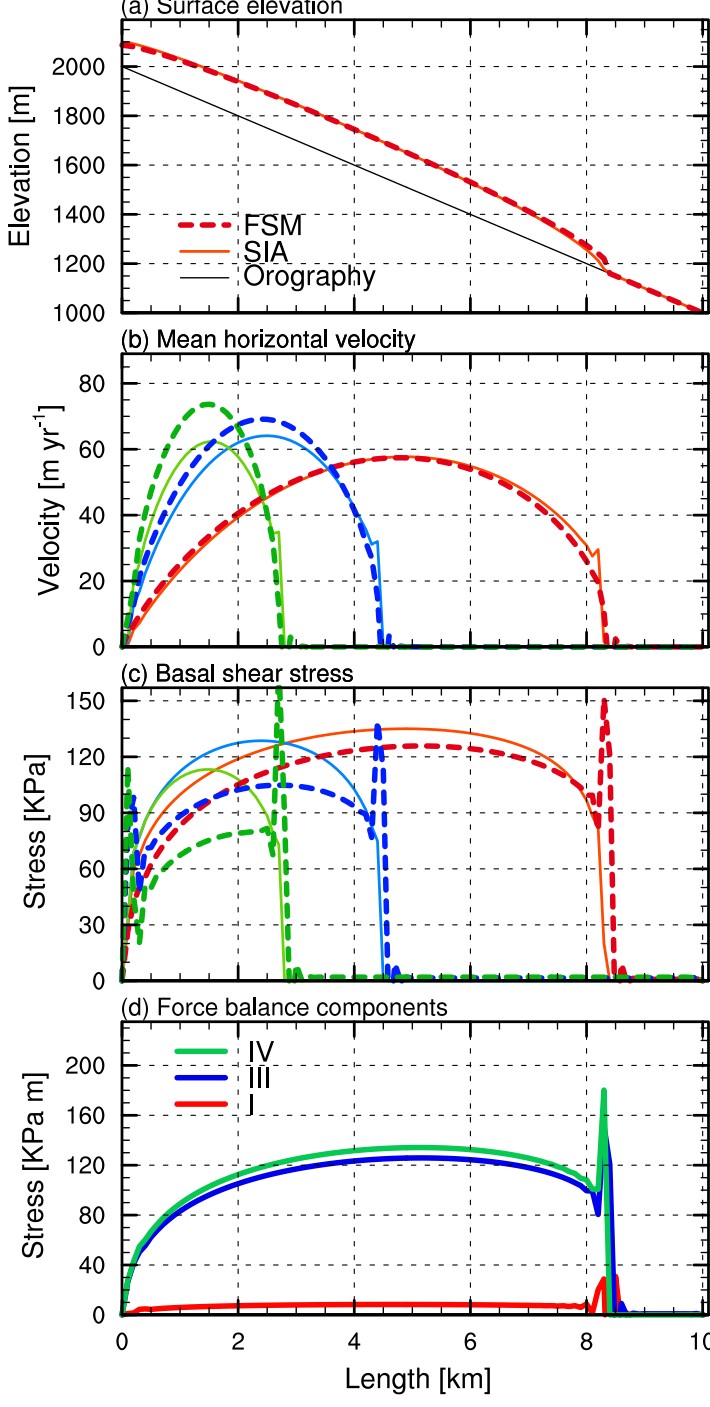

**Figure 5.** Steady state (a) surface elevation profile, (b) basal shear stress and (c) vertically averaged horizontal ice velocity calculated using SIA (solid line) and FSM (dashed line) for the large (red), medium (blue) and small glacier (green) configuration. The glaciers rest on a bed with a constant slope and a uniform width. (d) Force balance components in steady state for the large glacier configuration simulated using FSM. The green line (IV) is the driving force, blue line is the basal drag (III) and red line is the longitudinal stress gradient (I).




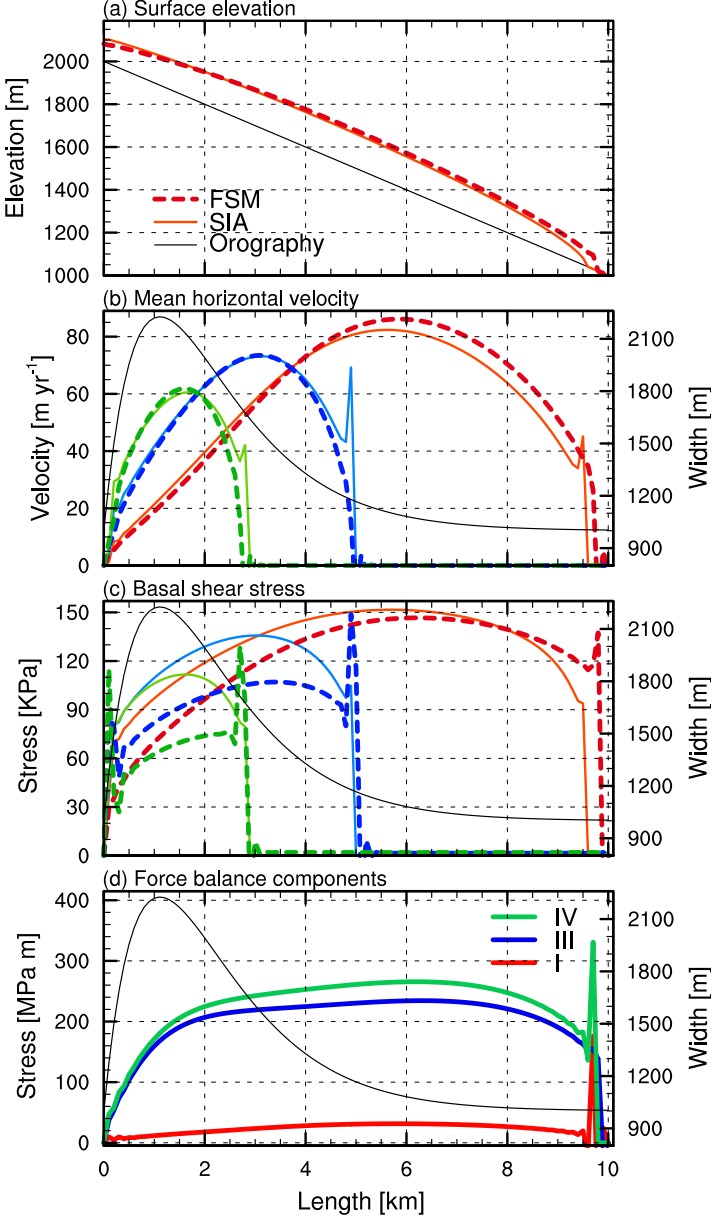

**Figure 6.** Steady state (a) surface elevation profile, (b) basal shear stress and (c) vertically averaged horizontal ice velocity calculated using SIA (solid line) and FSM (dashed line) for the large (red), medium (blue) and small glacier (green) configuration. The glaciers rest on a bed with a constant slope and a longitudinally varying width (depicted in the right axes of the figures in black). (d) Force balance components in steady state for the large glacier configuration simulated using FSM. The green line (IV) is the driving force, blue line is the basal drag (III) and red line is the longitudinal stress gradient (I).





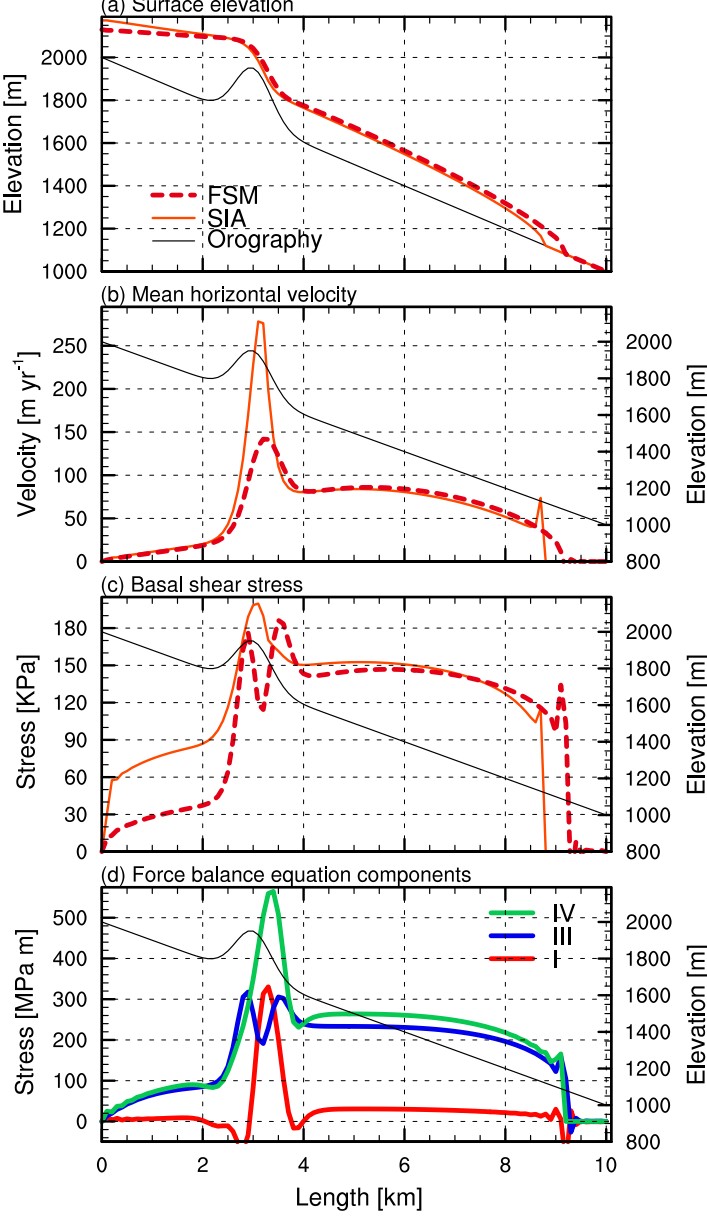

**Figure 7.** Steady state (a) surface elevation profile, (b) basal shear stress and (c) vertically averaged horizontal ice velocity calculated using SIA (solid line) and FSM (dashed line) for the large glacier configuration. The glacier rests on a bed with an imposed Gaussian bump (depicted in the right axes of the figures in black) and a longitudinally varying width. (d) Force balance components in steady state for the large glacier configuration simulated using FSM. The green line (IV) is the driving force, blue line is the basal drag (III) and red line is the longitudinal stress gradient (I).




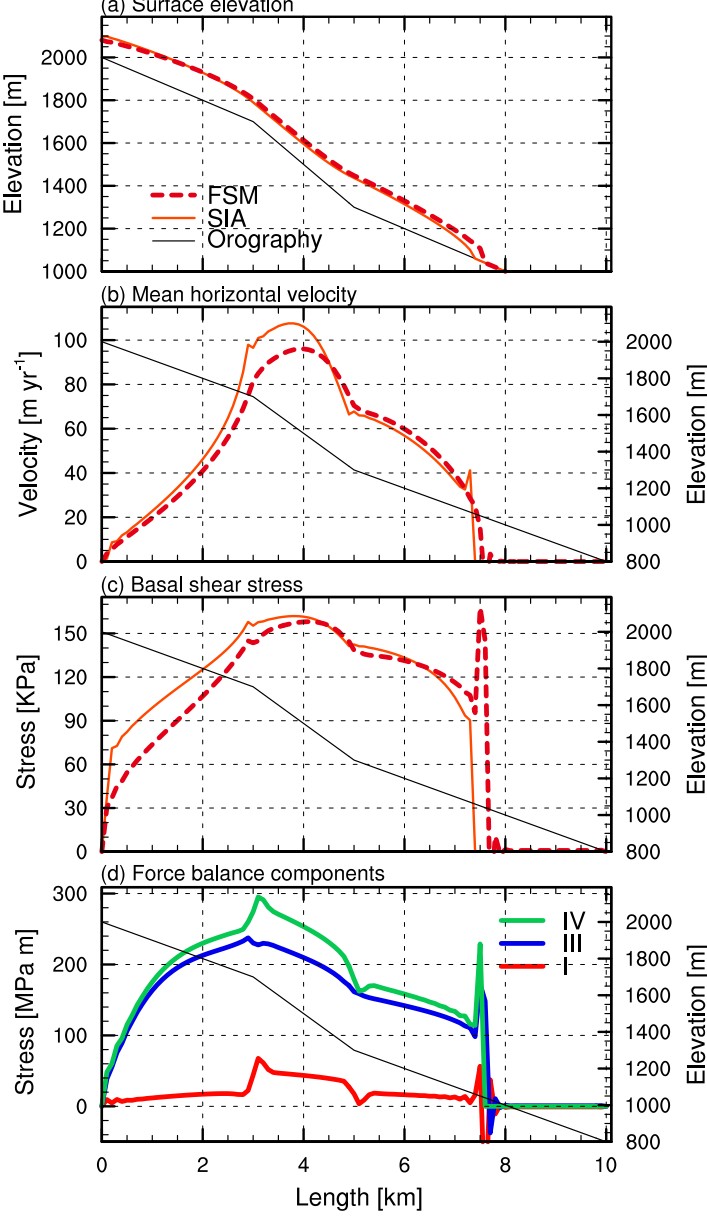

**Figure 8.** Steady state (a) surface elevation profile, (b) basal shear stress and (c) vertically averaged horizontal ice velocity calculated using SIA (solid line) and FSM (dashed line) for the large glacier configuration. The glacier rests on a bed with an imposed ice-fall (depicted in the right axes of the figures in black) and a longitudinally varying width. (d) Force balance components in steady state for the large glacier configuration simulated using FSM. The green line (IV) is the driving force, blue line is the basal drag (III) and red line is the longitudinal stress gradient (I).





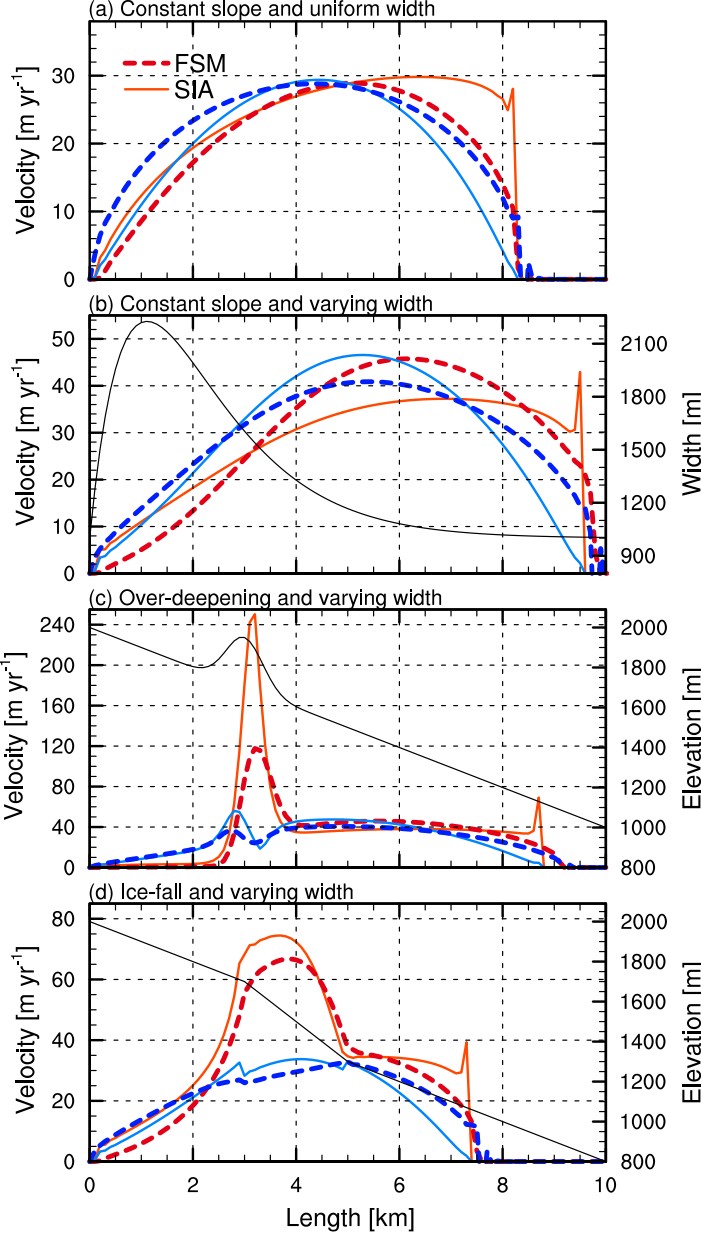

**Figure 9.** Steady state sliding (red) and deformation velocity (blue) for the large glacier configuration resting on a bed with (a) a constant slope and a uniform width, (b) a constant slope and a longitudinally varying width, (c) an over-deepening and a longitudinally varying width and (d) an ice-fall and a longitudinally varying width. The results are obtained using SIA (solid line) and FSM (dashed line). The glacier orography and width is depicted in the right axes of the figures.