# Peer review of "Numerical simulations of glacier evolution performed using flow-line models of varying complexity"

_Geoscientific Model Development, 2017_

## Referee Comment (RC1) · Anonymous Referee #1 · 26 Apr 2017

In this paper a comparison between the Shallow Ice Approximation (SIA) and a Full-Stokes Model (FSM) implemented in Elmer/Ice is presented. Results from several test-cases are shown and discussed and some conclusions are drawn from these. However, not much theory is presented that supports the results obtained. The paper is well-written but lacks important references and reflections on how the current findings relate to the research front line.

What is perhaps the most problematic issue with this paper is that it lacks references to [1] and [2] that deal with comparisons between SIA, Second Order Shallow Ice Approximation (SOSIA) and FSM. Moreover, in these papers the validity of the assumptions behind SIA and SOSIA is thouroughly investigated, theoretically and numerically. Also, the importance of the aspect ratio e – the characteristic height of the ice divided by the characteristic length – was numerically and theoretically studied. In order to merit

for publication, the authors must relate their results with respect to these papers. How much is actually new here and how much can be deduced from the results in [1] and [2]?

More detailed comments include:

- Page 3, line 19: At the. . .

- Presentation of bedrocks: please illustrate with one or more figures. As it is now it is difficult to understand the difference in the geometries of the simulations run in e.g. Sections 4.1 and 4.2.

- Overall, the references are fairly old. Apart from [1] and [2] it should be possible to include more relevant and up-to-date references.

- In most figures there is a mistake in the caption between (b) and (c).

To summarize, my opinion is that if the authors rewrite the paper thoroughly in relation to [1] and [2], it might be acceptable for publication. However, that strongly depends on how much is found to be new in the present paper when this is included.

References:

[1] Josefin Ahlkrona, Nina Kirchner, Per Lötstedt; A numerical study of scaling relations for non-newtonian thin-film flows with applications in ice sheet modelling. /Q J Mechanics Appl Math/ 2013.

[2] Josefin Ahlkrona, Nina Kirchner, Per Lötstedt; Accuracy of the zeroth-and second-order shallow-ice approximation–numerical and theoretical results. /Geoscientific Model Development/, 2013.

---

## Referee Comment (RC2) · Anonymous Referee #2 · 2 May 2017

General comments :

My main concern with this paper comes from the fact that we do not really see what is the message the authors want to pass. The paper simply compares results of 2 models of varying complexities under various contexts with regard to forcing scenarios and glacier/bed geometries, but no real conclusions are drawn as to which model to use under which circumstances. This is actually what the modeling comunity expects to learn from such a paper.

In that respect, the first phrase of the abstract should read 'Results of two...' instead of 'The performance of two..' because no discussion in terms of performance is actually proposed. Doing so would require comparing results in terms of computer efficiency (CPU), easiness of coding and/or validity of the results. I admit this last aspect is difficult to properly assess without studying real cases, still, glacier behaviours in response to climatic scenarios have sufficiently been investigated to have an idea on what is realstic or not (for instance the response lag as a function of the size of the glacier, the bed slope....). Moreover, given the degree of sophistication and the proven robustness of the Elmer model as correctly mentionned in the manuscript (reference to Pattn et al., 2008) it seems to me acceptable to consider the FS model as a realistic reference to which SIA results should be compared and to consequently derive a criterion for the correctness of the SIA simulations. These aspects have to be (at least partially) addressed for the paper to gain in interest and in my view to deserve publication.

I also agrree with referee #1 on the fact that the SIA approach relies on an asymptotic theory based on the aspect ratio.... proper reference should be made to this theory which dates back to quite some time ago ( eg, Hutter, 1983, Hindmarsh, 2004) and was considered again later within the isotropic/anisotropic duality and for newtonian rheologies (Mangeney and Califano, 1998) from which interesting results can also be exploited.

In any case, results should be discussed in terms of the aspect ratio (eg. Pattyn et al., 2008) bacause the validity of the SIA asymptotic theories (be it of the zeroth or second order) essentially relies on the value for this term which results mainly from the actual glacier geometry (typical thickness to length ratio, slope of the bedrock) and to a lesser degree from the intensity of the basal drag. This is the main reason why SIA has been abundantly used for modelilng grounded ice sheets for which this aspect ratio is generally small enough. One big issue for mountain glaciers modeling is whether or not the SIA is still applicable, and if so, under which conditions with regards to the glacier settings as given by its aspect ratio.

Moreover, as can be seen from the comments below, the paper as it is written suffers too many inconsistencies and errors which gives the unpleasant feeling of a botched job not properly checked before submission.

More specific comments follow :

- page3 line 8 : '..which depends on the local surface
slope and a ice thickness..' ->  supress the 'a'

- page 5 line 16 : '...under a small period random forcing and under..' -> '..period of ramdom forcing ..' ?

- page 6 line 10 : 'After the equilibrium states of three…',  'equilibrium state' is not the proper term.. we are here talking about steady state of glacier geometries which is not a true equilibrium state
-> please change

- Page 6 line 16 'For better clarity, the glacier length is shifted by ±1 km along the y-axis for the large and small glacier'. I am sorry but if the simulations start with zero volumes (assuming zero initial lengths) the shifts as represented on figure 1b is not correct.. I rather see a 2-km shift

- Results of the time dependent behaviour following the steadystate simulations (page 7 and figs 1 and 2 ) :
When results of fig 2 are commented, it is said : '...we can see that the rate of advance and retreat differs in our models with SIA producing shorter advance and retreat compared to FSM..' ... Honestly, when looking at the figure, I do not see any significant difference .. therefore the following conclusion does not hold anymore. This has to be changed.

- Eq. 11 : The formulation is not clear .. is it meant that TauV is the time after which the volume V reaches the proposed value after the equal sign ? If so , please state it differentlt, as it is written it is not a proper equation and it is anyhow confusing.

- page 7 line 18 : ' Unlike in Leysinger-Vieli and Gudmundsson (2004), whose full-Stokes model did not show consistent lag or lead in comparison to their SIA model, here, the response time in FSM is longer than the response time in SIA for all glaciers.' Why is there no attempt to discuss this contradiction with previous results ?  This illustrates my more general comments according to which there is an obvious lack of discussion of the proposed results. This way of achieving the paragraph is abrupt and frustrating for the reader.

- page 7 line 29 'The opposite is true for glacier volume that reacts faster in FSM compared to SIA. Also, as already seen, it is notable that glacier volume and length amplitudes are not equal in the two models.' .. again, I find these conclusions extremely drastic when actually looking at the correponding figures. I am sorry to say that the lag between SIA and FSM is almost insignificant on figure 3a.  (it is confirmed by phase differences of 53.1 and 56.5 degrees for FSM and SIA in table 3). As for the amplitudes , especially for volumes, they are almost the same ... I see a max difference of 0.2 km3 for volumes of around 10 km3....it makes a 2 % difference at the most ! not enough to assert that they are different in my view.

- Page 8  line 3 : ' Also, the phase lag increases with increasing frequency of the climatic forcing, possibly as a result of grid discretization since when a grid resolution is increased from 100 m to 20 m the phase lag decreases for almost 50%'

This is not a satisfactory explanation... if the authors suspect a model artefact, they should then try to solve for it. In the present case, before drawing any dubious conclusions, a sensitivity test on the grid size should be carried out until a convergence in the obtained phase lag is obtained. This is compulsory for the modeled time lag to become reliable and suitable for any form of interpretation. Either these simulations on the time lag as presented here should be removed or new simulations with an optimized grid size are proposed instead.

- page 8 line 11 : 'First, a uniform noise with a standard deviation of 100 m is superposed on E. Second, a forcing function derived form the Central European mean annual temperature record

(Luterbacher et al., 2004) is used' . It is absolutely necessary to show these two forcing curves especially when later the response of the glacier to the varibility in these fields is commented.

- page 8 line 19 'This is not the case for glacier volume evolution, where SIA lags FSM (see e.g., years 760-840 on Fig. 4d).' Sorry again... not observable from the figure ..moreover fig 4d does not represent glacier volume evolution ... has the paper been checked before submission ?

- page 8  line 19 : Additionally, it is notable that glacier length evolution in
FSM does not respond to some smaller short-term variability (e.g., year 800-900 and 1100-1200 on Fig. 4b)' ... what is the short term variability .. that of the forcing ? we have no means to assess it.

- page 8 line 20 : ' Figures 4c and 4d indicate a periodic reaction of glacier length and volume to year-to-year variability in the temperature record. The power spectrum (not shown) also shows high energy at a frequency of 250 years that seems to reflect on the glacier evolution.'  This is of no use if we do not see how the temperature record changes through time.

Page 9 line 1 : 'In Sect. 4.1 the glacier width W is kept uniform and it can be excluded from the equation', I do not see how the fact that W is uniform implies giving it up in Eq. 12

page 9  line 4 : Paragraph title says 'Constant bed slope and exponential change in glacier bed width' whereas the corresponding figure 5 talks about a bed with a uniform width…… → please check on that

- page 10 line 17 : 'Here, and in the following sections, the force balance components shown in Fig. 6d include the varying bed width. Applying this varying width leads to increase in the normal stress (compression) where the width decreases, and to decrease in normal stress (extension) where the width increases.' : This attempt to explain results of fig. 6d is a bit misleading. Indeed, warying widths imply changes in the normal stress , but the quantity as represented in the figure is the stress gradient which is not the same. This is confirmed by the fact that on the figure we do not see any obvious link between the stress gradient and the glacier width. The interpretation (if there is one of interest) should then be reconsidered or removed as it is confusing in its present state.

- Figure 7 : There is an inconsistency between the caption and the figure. (b) is the vertically averaged horizontal velocity and (c) is the basal shear stress.

- Last paragraph section 4.2 : The proposed interpretation of figure 7 is by far too vague and approximative :

       -'At the length of about 3000 m, or just upstream of the bed bump...', The distance of 3000 m matches the bump top not its upsream part -> please be more precise

       -'Above and just downstream of the bump, the driving stress reaches its maximum due
to the large surface slope' .. when mentioning 'above and just downstream of the bump', are the authors trying to explain the double peak in the basal shear stress ? There is obviously a singularity here in the SIA derived basal stress which can not be solely explained by the surface slope as stated. If it were the case, the surface slope should undergo a similar double peak and the local minimum in basal stress should not correspond to approximatively the maximum in surface slope as on the

figure. One possible explanation could reside in the double dependency of the SIA expression of the driving stress (proportional to both the ice thickness and the surface slope), although I am a bit dubious on that in the present case owing to the strong singularity in the shear stress. I would rather suspect a deficiency in the SIA representation in this specific context.

This brings me to a more general comment :

We are probably here addressing the most interesting part of the paper, namely the local degradation of the aspect ratio due to a singularity in the basal topography. This aspect has already been abundantly addressed in the literature and authors should reconsider their interpretation by referring to appropriate papers. Above all, authors should interpret their results in terms of the aspect ratio (as also suggested by reviewer 1). Here we obviously face a situation where locally fast varying slopes degrade the aspect ratio making SIA representations strongly deviate from reality. This is typically the kind of situation where FSM approaches should make up for the SIA deficiencies. This aspect is fundamental but is obviously missing here and a significant rewriting of the paper should then be considered.

- Fig 8 : again a confusion similar to that of fig 7 in the caption. Making copy-paste captions from one figure to the other can be a good way of saving time provided a minimum check is done.

- Ice fall result interpretation : Here also, although maybe to a lesser degree, the bedrock perturbation leads to changes in the SIA and FSM representations which should be interpreted similarly as suggested above.

- page 12 line 24 : 'For a constant bed slope and a uniform glacier width, the sliding velocity is higher in SIA compared to FSM in the area of high ablation. This supports our conclusion that in SIA glaciers flow faster to the area of higher ablation, which in turn makes the volume response in SIA slower compared to FSM'
I do not understand this statement... if with the SIA glaciers flow faster to areas of higher ablation, it induces a higher turn over which in my view (unless I am wrong) should lead to faster response times at least when the equilibrium altitude increases. The simulation is for a constant bed slope and a uniform glacier width, corresponding to figure 2 and Table 2 where inferred volume response times are smaller with SIA compared to FSM .... how can authors talk about 'volume response in SIA slower compared to FSM' then ?

- Page 12 line 30 : 'This does not necessarily mean that the highest surface velocity simulated using SIA of about 250 m year $-1$ is overestimated. Wangensteen et al. (2006) showed that the highest surface velocity measured on Nigardsbreen (Norway) can reach 489 m year $-1$ at the main ice-fall, which is almost twice as high than the highest velocity obtained in this study'
This comparison seems to me irrelevant as an attempt to explain the high velocities of the SIA model. Comparing a real case glacier with a theoretical glacier as proposed in this study does not make sense as the settings differ notably. These high velocities should rather be interpreted as the deficiency of the SIA approach in the present case. Many previous studies have shown that for mountain glaciers, if SIA representations remain usually quite realistic for volume changes, on the other hand they often yield irrealistic representations of the velocity field as correctly stated page 2 line 16.

References

Hutter, K.: Theoretical Glaciology: material science of ice and the mechanics of glaciers
30 and ice sheets., D. Reidel Publishing Company, Terra Scientific Publishing Company
(ISBN90277 1473 8), 1983.

Mangeney, A. and Califano, F (1998) The shallow ice approximation for anistropic ice: formulation
and limits, J. Geophys. Res. 103(B1) 691-705

Hindmarsh, R.C. A. 2004 : A numerical comparison of approximations to the
Stokes equations used in ice sheet and glacier
modeling,    JOURNAL    OF    GEOPHYSICAL    RESEARCH,    VOL.    109,    F01012,
doi:10.1029/2003JF000065, 2004

---

## Referee Comment (RC3) · G. J.-M. C. Leysinger Vieli (Referee) · 2 May 2017

General comments: The authors compare results from a shallow ice approximation (SIA) model with a full-Stokes (FS) model (FSM), which uses ElmerIce. Several model experiments are performed to assess model differences in glacier evolution. The experiments all start from a steady-state that has been chosen for both models as to represent the same initial steady-state geometry. The experiments include a step-wise forcing as well as periodic fluctuations of the equilibrium line altitude. Further a change in glacier width and changes in bed slope are introduced. The force balance equation is used on the FSM to gain insights into which components of the force balance are important along glacier for the various experiments.

The paper is for the largest parts carefully written but in my view too long and it becomes not clear what the authors want to get out of it. While reading I got rather confused as it is not always clear if the comparisons between model results are comparable. The model set up is e.g. not the same as the one used in Leysinger Vieli and Gudmundsson (2004) (e.g. using no-slip) but the results are much compared with each other and differences mentioned but the possible reason behind it is neither analysed nor discussed. I did not understand the aim of this model comparison study and neither did I understand what can be learned from it. It is not clear if some general recommendations can be made that are valid for all SIA and FS models.

The main weakness is that the authors compare the two models that have been adjusted by some parameters to produce the same initial geometry but the effect of this adjustment on the calculations is not really tested and accounted for. A further weakness is the grid resolution. It becomes not clear in the paper if the grid resolution is correct. It does not mean that the FSM needs the same resolution as the SIA - especially at the front the resolution might be too coarse.

I am not sure what the paper is adding to the current knowledge, what is new to previous studies. If the paper could be more specific in saying for what this comparison is made and something new is learned from it, the contribution would be valuable.

Specific comments: In order to obtain the same steady state geometry for the SIA and FS model, the parameters for sliding and deformation have been tuned for the SIA model. For the three different slope three different geometries are obtained that are initially the same for both the SIA and the FS model. The models allow for sliding at the base described by a Weertman-type sliding law. The sliding parameter in the FSM is connected to the sliding parameter in the SIA model so that the sliding velocities are equal in both models. This is done by choosing the mean ice thickness and divide it by the sliding parameter used in the SIA model. Reading equation (2) I understand that for the sliding velocity the thickness varies along the glacier. But for the FSM in equation (4) using the sliding parameter C as defined in equation (5) the thickness H is the mean thickness. So H does not change along glacier. This is a problem at the front

where the ice thickness becomes very small. At the front the basal velocity in FSM and the sliding velocity in SIA are not the same due to not accounting for the change in thickness in the FS model. This effect is seen in the basal shear stress as well as in the force balance components (e.g. Figures 5-8 in (c-d)), where a high peak in the FSM is obtained at the glacier front. This might also be the reason why the SIA seems more responsive than the FS model (e.g. Figure 2).

As a very first experiment and to test the two models (and not the physics) I would have expected to run the steady state using an x-dependent mass balance distribution and therefore without any altitude mass balance feedback. The steady state front positions must be equal for both models and can be determined analytically allowing to estimate the accuracy of the two numerical models and also determine the most suitable grid resolution for the two models.

For the comparisons with Leysinger Vieli and Gudmundsson (2004) it is not clear what the observed differences mean - neither has been considered that the front evolution in the FS model used by Leysinger Vieli and Gudmundsson (2004) is not bound to spatially fixed grid points as it uses and adaptive grid moving with the surface. The authors of this study mention on page 8 that a smaller grid leads to a 50% change in phase lag. This makes me wonder if the results in general have a grid dependency, especially at the front. The changes in length do seem to be contained within 1 grid size (e.g. Figures 2 (b), 3 (b) and 4 (b and d), which is a random and not a significant result. However this is not reflected in the discussion of the results. This grid difference is also important for the discussion of advance and retreat rates. Looking at Figure 2(b) the slope of the curve for an advance looks slightly different between SIA and FS model but nearly identical for the retreat. Furthermore, the statement made in the discussion (p. 13 lines 6-11) on steady state length for an advance and a retreat does not take into account that the current paper (Rimac and others) uses a linear mass balance function whereas Leysinger Vieli and Gudmundsson (2004)use a non-linear mass balance function (two different gradients above and below the ELA). A

further difference is that Leysinger Vieli and Gudmundsson (2004) used a no-slip basal condition.

For the experiments using a varying width, I am not sure what we learn from it for the three different glaciers. The width varies along flow and not relative to the glacier length. This is fine comparing the models but what does it mean for inter comparison of different sized glaciers? Why continue all experiments with the change in width instead of looking at just one added complication.

Technical corrections: P1, L8: Another paper to consider is M. P. Lüthi, 2009. Transient response of idealised glaciers to climate variations, Journal of Glaciology, Vol. 55, No. 193.

P1, L20: 'Leysinger Vieli' without hyphen.

P2, L16/17: Where is the velocity so different? Front? And when? In steady-state?

P2, L19: What do you mean by 'crudely studied'? Explain!

P2, L20: Complexity is used in both sentences but I believe sth. else is meant. Elaborate - be more precise.

P2, L31: e.g. Lüthi, 2009 applied a sinusoidal variations of the ELA to investigate the response in glacier length and volume.

P3, L8: 'and the ice thickness' instead of 'a'.

P3, L14/15: I find this an odd argument. You can still get the numerics wrong even when others had correct results (e.g. grid size etc.). You need to test your specific case.

P3, L19: 'At the lower boundary' instead of 'he lower'. Because of Weertman type friction law I know you mean the base - but you might want to make this clearer which lower boundary (downstream end or base?) you mean.

P3, L32: I believe that 100m might be to coarse at front for the FSM. Did you check for the Courant-Friedrichs-Lewy stability condition? abs((u*delta_t)/delta_x))<= 1 using the largest velocity u in the system.

P4, L4: instead of 'tau to the power of 3' would not '1/m' instead of 3 be more correct?

P4, L4: f_d and f_s introduce a different rheology to the glaciers. Does this make sense?

P4, L6-8: Sentence is confusing as it is not clear if the difference is between SIA and FSM (which would be correct) or between the small, medium and large glaciers.

P4, L23: This condition makes that H is constant but the same is not true in Equation (3) where H varies along the glacier. This leads to large differences where thickness changes (e.g. at the front)!

P4, L28: Give the slopes also in degrees (angle).

P5, L8: You use 'Second' - where is 'First'?

P6, L4: Does it make sense to vary the slope by the same amount in slope? This means that the steeper part is relatively steeper for the low slope than for a higher slope. Why not the same slope for the steeper part? Or the same relative increase?

P6, L17: Not clear in Figure 1 what the length is. Where are these +- 1km?

P6, L19: SIA is faster due to sliding factor f_s/H (C) - see main comments.

P6, L21/22: What is the measure for steady-state?

P6, L26/25: Not sure your aim is possible with f_s and f_d parameters that are different in the models.

P7, L5/6: It differs for advance and retreat - but does it really differ between the models? At least for the retreat it does not look so. See main comment on grid size.

P7, L18: In Figure 2 a 'slight' change is seen for advance and 'none' for retreat. Front

might be the reason - grid resolution as well as f_s and f_d.

P7, L28: 'glacier length in SIA reacts faster' this again might depend on fron and chosen f_s.

P8, L18-22: This interpretation is an over interpretation of data within 1 grid cell. It's a resolution problem. Within one grid cell no statement can be made.

P9, L25-33: This paragraph is not clear to me. The problem here is C in the FSM. But are you not comparing apples with pears as all three sized glaciers have different f_s?

P10, L15/16: Does it make sense to compare the velocities for the different glaciers? The smallest glacier is sitting nearly entirely in the widest part but for the large glacier this is only a small region compared to its total volume. Should the width be chosen the same for all or relatively varying in width at the same position for all (e.g. wide in the first third)?

P10, L23: 'non-uniform glacier width' I did not understand what the change is to the previous 'exponential change' is. Equation is the same? I believe it's the same change in width?

P11, L1: add 'end' to make it clearer - 'at the bump's downstream end'.

P11, L3-9: Paragraph not very clear - not always clear if you are speaking of SIA or FSM!

P12, L19: How much does your statement depend on the chosen width?

P12, L30-33: Velocity depends on thickness and slope - is your glacier comparable to Nigardsbreen? What do you want to say by this statement?

P13, L4: 200 meters is only 2 times the grid size - this is not 'large'.

P13, L8: 'shorter than 5 km' - only one small glacier was studied with the definition of small ranging between 1<= length <= 5km. But one can not make a general statement

of smaller than 5km from the Leysinger Vieli and Gudmundsson (2004) study - rather on aspect ratio.

P13, L8: 'a SIA model' instead of 'an'.

P13, L9: it's not so much 'length' but more importantly it depends on the aspect ratio of the glacier - so a small aspect ratio might not be well represented by SIA

P13, L10: How different in length is 'different' here?

P13, L14: Statement not very clear here (grammatically wrong?).

P15, L36: 'Gagliardini' and not 'Gqgli..'

P17, Table2: Not clear if for an advance or retreat!

P18, Figure 1b: Not clear how long the glacier is. Here they seem shifted by 2 km.

P19, Figure 2: Why do they not start from the steady-state position? Show the start and explain what it is.

P20, Figure 3b: the length change in the minimum seems to be rather due to the grid resolution - this has then also an effect for the Maximum (two grid cells there (added to the first difference).

P21, Figure 4: Again grid size differences!

P22, Figure 5c,d: peak in FSM due to C (or rather matching it to f_s with a constant ice thickness H).

P22-25, Figures: Caption to (b) and (c) are swapped in the text.

---

## Author Comment (AC1) · 5 May 2017

**Reviewer 1:**

We would like to thank Reviewer 1 on his/her comments. Our answers are given as following:

General comments:

We believe that we failed to stress the guiding idea of our paper. This is visible from reviewers' comments where he/she misses the reflections of the current findings to the research front line. Hereby, we would like to make a comment on that statement.

Projecting the future behavior of a valley glacier can only be done when a careful calibration with the past glacier record is done. This is the best way to assure that the initial state of a model integration is realistic. In fact, the choice of initial state determines the outcome to a large extent. Calibration with a historical record (normally glacier length) is best done by reconstructing the past mass balance forcing (e.g., Oerlemans, 1997). This implies the use of a control method, in which mass balance parameters have to be optimized to make the mismatch between observed and simulated glacier length as small as possible. For instance, if an equilibrium-line altitude (ELA) is adjusted every five years for a 200-year record, 40 parameters (values for the ELA) have to be found. This requires a few hundred runs with a dynamic glacier model. If one wants to apply this procedure to a set of glaciers, thousands of model runs have to be done.

It is clear that full-Stokes models (FSM) are too computer-time consuming in order to perform the defined simulations. Models based on the Shallow Ice Approximation (SIA) however, require several orders-of-magnitude less computer time, and are therefore more suitable for control simulations. For example, the computing time in SIA is less than a minute while for FSM is about an hour for a simple simulation of 500 years (for the present simulations, but also shown by LeMeur et al., 2004 and Schäfer et al., 2008).

Models based on the SIA capture most of the broad characteristics of valley glaciers, and therefore may be a good candidate for the type of numerical experiment described above (i.e., numerical experiment with a focus on complete picture of historic climatic variation). Most mountain glaciers are located in a mass balance field where the balance rate increases gradually with height. The landscape hypsometry then determines the equilibrium extent of a glacier. Ice mechanics set in because the thickness of a glacier depends on its size, and feeds back on the balance rate by the corresponding change in surface elevation. This implies that a dynamic model should first of all deliver the relation between glacier size (in terms of length) and mean ice thickness. For reasonable smooth topographies, it seems that SIA-models can do that well (e.g., Oerlemans 1986, 1997), even if some of the details of the ice mechanics are better represented in full-Stokes models.

In this paper, we investigate whether these ideas hold. We compare runs performed with an SIA model with runs of a full-Stokes model (FSM based on the Elmer/Ice code). We focus on the response of bulk glacier characteristics (length and volume) to different climatic forcings. Although there are studies examining general differences between SIA and FSM based on a single forcing function and one glacier bed profile (e.g., Pattyn, 2002 and

Leysinger Vieli and Gudmundsson, 2004), a study that systematically builds up the complexity of the defined problem by applying several configurations of climatic forcing and glacier bed characteristics has not been performed up to our knowledge. Also, we derive and test an equation (Equation 1. in the paper) that allows users of Elmer/Ice code to study glaciers in 2D simulations when glacier width is included. This equation is of a great importance because Elmer/Ice code does not have developed solver that accounts for changing glacier width in 2D set-up.

We acknowledge the reviewers' concern about missing to relate our results to the characteristic aspect ratio. Again, we repeat that the guiding idea of the present study is to examine the response of glacier length and volume to different climatic forcings. This can be done only if the simulations in two models start from the same steady state (that in this case becomes the new initial condition). In this new initial condition, the aspect ratio for all glaciers is <0.01. Alkhrona et al. (2013) argue that aspect ratio of 0.1 sets the limit of validity for SIA. This gives a credit to our simulations of glacier evolution under different climatic conditions. Moreover, having the computing time in mind, SIA simulations provide valuable results against which we can test the simulations from FSM model.

As the reviewer's comments are mainly focusing on the technical details of the study, we would like to emphasize one more time the main point of the paper: that the used FSM model shows consistent lag in climate simulations, an important message we try to transfer. This raises a question if a sophisticated ice-flow model, such as the one based on Elmer/Ice code, is capable of correctly simulating a response time of a real mountain glacier or is a simple model based on SIA more suitable for climate simulations (as we stated in the discussion section).

More detailed comments:

*P.3, L.19:* The comment will be included.

*Presentation of bedrocks:* In the present paper, we have included the equations used to shape the bedrock, but, for better clarity we will add a figure illustrating the bedrock (see the attached figure 1). Please note that the main difference between 4.1 and 4.2 is the glacier width. As already explained it in Section 2.2 (P.5, L.22-25), we first study the simple glacier with a uniform width that rests on a bed with a constant slope and second, that we study the glacier with an exponentially varying width that rests on a bed with a constant slope. We noticed a mistake in the title of Section 4.1 that probably lead to the confusion. The correct title should be: Constant bed slope and uniform glacier width.

*References:* We have overseen some important references, as correctly concluded by the reviewer. Newer references will be added.

*Figures:* The comment will be included.

[Figure]

Figure 1. Model set-up showing three different glacier bed topographies. Note that red and green line are shifted along y-axis for 200 m.

---

## Author Comment (AC2) · 9 May 2017

**Reviewer 3:**

We would like to thank Reviewer 3 (dr. Leysinger Vieli) for her comments. Since the reviewer sent a lengthy letter hereby we will orient ourselves on her general and specific comments. We will take into account all of the technical comments that do not require discussing (without specifically addressing them in this reply), while the rest (i.e., the concern on the sliding parameters and grid resolution) are already covered in general and specific comments.

General and specific comments:

First, as it is the case with the first two reviewers, we feel that we need to better stress the novelty of our paper. Here, we will summarize our guiding idea and the main conclusion, but for detailed reading we refer the reviewer to our response to Reviewer 1.

Models based on the SIA capture most of the broad characteristics of valley glaciers, and therefore may be a good candidate for the numerical experiments in which future behavior of a valley glacier is studied. In these type of experiments, careful calibration with a historical record is a necessity. Since such experiments are computer-time costly, models based on SIA are good candidates to test the results of a full-Stokes model.

In this paper, we investigate whether these ideas hold. We compare runs performed with an SIA model with runs of a full-Stokes model (FSM based on the Elmer/Ice code). We focus on the response of bulk glacier characteristics (length and volume) to different climatic forcings. Although there are studies examining general differences between SIA and FSM based on a single forcing function and a simple glacier bed profile (e.g., Pattyn, 2002 and Leysinger Vieli and Gudmundsson, 2004), a study that systematically builds up the complexity of the defined problem by applying several configurations of climatic forcing and glacier bed characteristics has not been performed up to our knowledge. Additionally, we derive and test an equation (Equation 1 in the paper) that allows users of Elmer/Ice code to study glaciers in 2D simulations when glacier width is included. This equation is of great importance because Elmer/Ice code does not have a developed solver that accounts for changing glacier width in 2D set-up.

As the reviewer's comments are mainly focused on the technical details of the study, we would like to emphasize one more time the main point of the paper: that the used FSM model shows consistent lag in climate simulations, an important message we try to transfer. This raises a question if a sophisticated ice-flow model, such as the one based on Elmer/Ice code, is capable of correctly simulating a response time of a real mountain glacier or is a simple model based on SIA more suitable for climate simulations (as we stated in the discussion section).

Second, as the main weakness the reviewer claims that we adjust our SIA model by "some parameters to produce the same initial geometry but the effect of this adjustment has not been discussed and accounted for". As it can be seen from the reviewers' further comments, we believe she speaks of the sliding parameters. Allow us to clear up this misunderstanding. Both models use the Weertman-type of sliding law. To study mountain

glaciers, sliding has to be included (Leysinger Vieli and Gudmundsson, 2004 on the other hand exclude any basal motion, and they can only speculate how its inclusion would reflect on their results). In SIA, the sliding law is not modelled explicitly, but its bulk effect is absorbed in the sliding parameter that is included in the equation for sliding velocity (Equation 2 in the paper). The values suggested by Budd at al. (1979) are $1800 * 10^{-15}$ Pa$^{-3}$ m$^2$ yr$^{-1}$ for sliding and $0.06 * 10^{-15}$ Pa$^{-3}$ yr$^{-1}$ for deformation. These values are empirical constants and can be subject to some adjustments (Greuel, 1992). In FSM, the sliding law is presented through basal shear stress (Equation 4 in the paper) that is defined as a non-linear function of a basal (i.e., sliding) velocity and a sliding parameter. Elmer/Ice code manual *does not define* the sliding parameter but *only suggests* the possible value of 0.03 MPa m$^{-1/3}$ yr$^{1/3}$.

We need the specific model set-ups (e.g., initial state, boundary conditions and sliding law, time step and grid set-up) to be identical to make a realistic comparison between the simulations performed using different models. This means that the sliding parameters, although differently implemented in our models, must be as comparable as possible. Since Elmer/Ice code manual *does not define* the sliding parameter, we performed series of experiments to test different values for sliding (i.e., we performed large number of experiments using both SIA and FSM to obtain the correct values for sliding). Our experiments led us to conclusion that using the parameters defined in Table 1 of the paper we will obtain steady state length and volume that are equal in the two models for the three glaciers.

We wanted to test the realism of our defined sliding parameters (i.e., if the sliding parameters of the two models can be compared to each other). Thus, we hypothesized that sliding velocities in FSM and SIA are equal, and we derived an equation (Equation 5 in the paper) which shows that sliding parameter in FSM can be derived from the sliding parameter in SIA. Please note that the sliding parameters derived using this mathematical formulation differ by about 20% from the ones presented in Table 1. Nonetheless, the defined equation can serve as a guidance to readers how to correctly choose their sliding parameter when using Elmer/Ice code.

The reviewer further states: "At the front the basal velocity in FSM and the sliding velocity in SIA are not the same due to not accounting for the change in thickness in the FSM model. This effect is seen in basal shear stress as well as in the force balance components." This is not correct because the reviewer's conclusion is based on a misinterpretation of Equation 5. If we assume to have a no-slip basal condition in FSM, the peaks are still present (please see the attached Figure 1). In this figure, we plot steady state basal shear stress and horizontal ice velocity simulated using FSM for a no-slip boundary condition (red), a model set-up with high sliding coefficient (blue) and a model set-up with the sliding coefficient presented in the paper (green). We can see that high peaks at the glacier head remain in all set-ups. This means that the choice of sliding parameter does not influence the presence of this instability. What causes the instabilities is already discussed in the paper at P.9, L.29-34.

The reviewer misses the discussion on the effect of different sliding parameters on our results. In Section 4.1 we already present a good indication of the influence of changing sliding parameters on our results. There we explain the influence of sliding parameter on

the ice velocity and basal shear stress (paper P.9, L18-24). We do not discuss this influence on the instabilities seen on figures for the basal shear stress, since we could not find any connection. Nonetheless, in order to improve the paper following the reviewer's comment, we will elaborate the discussion in the revised paper.

Third, the reviewer is puzzled about the dependence of our results on the structured grid definition and grid resolution. In the present type of experiments, it is necessary that the grid is equally defined in both models. We do not believe that in detail comparison can be done if the grids are of different resolution or if the grid in FSM is adjustable at the glacier front. Since in these experiments we simulate glacier length (and volume) evolution, it is only natural that some of the results will be limited to one-to-two grid points, but more important is *the difference in response times* between our models (a result that the reviewer does not comment). To give better insight to the reviewers' question, we will investigate how the glacier time evolution under different climatic conditions changes when we use a higher grid resolution.

Fourth, the reviewer asks for more elaboration on the goal of the experiments with a varying glacier width. Please note that the glacier width, as defined in Equation 8, is defined for a large number of mountain glaciers that have a wide accumulation basin and a narrow tongue. Therefore, we do not see anything disputable in our formulation. As we already stated at the beginning of this letter we perform the experiments described in Section 4.2-4.4 of the paper to test an equation (Equation 1 in the paper) that allows users of Elmer/Ice code to study glaciers in 2D simulations when glacier width is included. Also, we extend our analysis by including more complicated bed profiles since many mountain glaciers have a bed with a reversed bed slope or ice-fall over a certain distance along the flow line.

Finally, throughout her letter, the reviewer points to too many differences in the model set-up between our study and the one from Leysinger Vieli and Gudmundsson (2004) to make a direct comparison. She states: "For the comparisons with Leysinger Vieli and Gudmundsson (2004) it is not clear what the observed differences mean …", and "The model set-up is e.g., not the same as the one used by Leysinger Vieli and Gudmundsson (2004) but the results are much compared with each other…". In order to address the reviewer's concern, we will no longer make in depth comparison of our results to Leysinger Vieli and Gudmundsson (2004) in the revised manuscript.

[Figure]

Figure 1. (a) Basal shear stress and (b) horizontal ice velocity simulated using FSM. Green line represents the result from the paper, red line is the simulation with no-slip condition and blue line is the simulation with high sliding.

---

## Author Comment (AC3) · 9 May 2017

**Reviewer 2:**

We would like to thank Reviewer 2 for his/her comments. Since the reviewer sent a lengthy letter hereby we will orient ourselves on his/her general comments. We will take seriously all of his/her specific comments that do not require additional discussing (e.g., suggested references, rewriting the manuscript in order to check the spelling and inconsistencies, rephrase the sentences/phrases pointed by the reviewer in order to make them clear). The rest of the comments are addressed separately as following:

General comments:

As it is the case with the other two reviewers, we feel that we need to better stress the novelty of our paper. Here, we will summarize our guiding idea and the main conclusion, but for detailed reading (i.e., in detail motivation, comment on the CPU time differences between the models, comment on the relation to the characteristic aspect ratio) we refer the reviewer to our response to Reviewer 1.

Models based on the SIA capture most of the broad characteristics of valley glaciers, and therefore may be a good candidate for the numerical experiments in which future behavior of a valley glacier is studied. In these type of experiments, careful calibration with a historical record is a necessity. Since such experiments are computer-time costly, models based on SIA are good candidates to test the results of a full-Stokes model.

In this paper, we investigate whether these ideas hold. We compare runs performed with an SIA model with runs of a full-Stokes model (FSM based on the Elmer/Ice code). We focus on the response of bulk glacier characteristics (length and volume) to different climatic forcings. Although there are studies examining general differences between SIA and FSM based on a single forcing function and a simple glacier bed profile (e.g., Pattyn, 2002 and Leysinger Vieli and Gudmundsson, 2004), a study that systematically builds up the complexity of the defined problem by applying several configurations of climatic forcing and glacier bed characteristics has not been performed up to our knowledge. Additionally, we derive and test an equation (Equation 1 in the paper) that allows users of Elmer/Ice code to study glaciers in 2D simulations when glacier width is included. This equation is of great importance because Elmer/Ice code does not have a developed solver that accounts for changing glacier width in 2D set-up.

As the reviewer's comments are mainly focused on the technical details of the study, we would like to emphasize one more time the main point of the paper: the used FSM model shows consistent lag in climate simulations. This raises the question whether a sophisticated ice-flow model, such as the one based on Elmer/Ice code, is capable of correctly simulating a response time of a real mountain glacier or whether a simple SIA model is sufficient (as we stated in the discussion section).

Specific comments:

*P.6, L.16:* Please note that the length for the large glacier is really shifted by 1 km (and for the small glacier by -1 km). What might be confusing is that it is difficult to notice the

correct volume for the glacier after 1 year of simulation. At that time, the volume is no longer 0 km$^3$ (as it might seem on the figure), but about $0.03*10^{-4}$ km$^3$. Having that in mind, we believe it is understandable that the length cannot be 0 km (since the calculation for the glacier length depends on the ice height and thus, any grid point that has the ice height >0 is taken in length calculation).

*Equation 11:* Tau$_V$ is the time at which the volume $V = V_2 - (V_2 - V_1)/e$.

*P.7, L.29, P.8, L.3 and P.12, L.24:* We would like to thank the reviewer for his/her concern. We will repeat our simulations using higher grid resolution in order to be more accurate in our discussion.

*P.8, L.11:* We believe that it is important to show the mentioned figures (especially the ones for the length evolution). In these figures, we can see that lengths in SIA and FSM do not respond equally to the forcings and that there is at least 10 years delay in the response of FSM compared to the response of SIA. We find that difference substantial.

*P.8, L.20:* For clarification, we will overlay a temperature record on the length evolution plot, the same as we have overlaid change in ELA on Figures 2-3.

*P.12, L.30:* We agree with the reviewer that the attempt to justify high velocity in SIA by bringing into discussion paper by Wangensteen et al. (2006) is unnecessary. We will incorporate the discussion suggested by the reviewer.